# Coenzyme A corrects pathological defects in human neurons of PANK2-associated neurodegeneration

Daniel I Orellana[1,†], Paolo Santambrogio[1,†], Alicia Rubio[2], Latefa Yekhlef[3], Cinzia Cancellieri[2], Sabrina Dusi[4], Serena G Giannelli[2], Paola Venco[4], Pietro G Mazzara[2], Anna Cozzi[1], Maurizio Ferrari[5,6], Barbara Garavaglia[4], Stefano Taverna[3], Valeria Tiranti[4], Vania Broccoli[2,7,‡] & Sonia Levi[1,6,*,‡]

## Abstract

Pantothenate kinase-associated neurodegeneration (PKAN) is an early onset and severely disabling neurodegenerative disease for which no therapy is available. PKAN is caused by mutations in *PANK2*, which encodes for the mitochondrial enzyme pantothenate kinase 2. Its function is to catalyze the first limiting step of Coenzyme A (CoA) biosynthesis. We generated induced pluripotent stem cells from PKAN patients and showed that their derived neurons exhibited premature death, increased ROS production, mitochondrial dysfunctions—including impairment of mitochondrial iron-dependent biosynthesis—and major membrane excitability defects. CoA supplementation prevented neuronal death and ROS formation by restoring mitochondrial and neuronal functionality. Our findings provide direct evidence that PANK2 malfunctioning is responsible for abnormal phenotypes in human neuronal cells and indicate CoA treatment as a possible therapeutic intervention.

**Keywords**  Coenzyme A; hiPSC; iron; neurodegeneration; PKAN
**Subject Categories**  Neuroscience; Stem Cells

## Introduction

Pantothenate kinase-associated neurodegeneration (PKAN; OMIM *606157) is an autosomal recessive movement disorder caused by mutations in *PANK2* (Zhou *et al*, 2001). It belongs to a heterogeneous group of neurodegenerative diseases, collectively known as neurodegeneration with brain iron accumulation (NBIA), which are characterized by severe iron overload in specific brain regions,

neurodegeneration and extrapyramidal dysfunction (Hayflick *et al*, 2006; Levi & Finazzi, 2014). Mutations in *PANK2* account for approximately 50% of NBIA cases in Caucasian populations (Colombelli *et al*, 2015). PKAN usually manifests in early childhood with gait disturbances and rapidly progresses to a severe movement deficit with dystonia, dysarthria, and dysphagia (Hartig *et al*, 2012). A distinguishing feature of this disease is the presence of the eye-of-the-tiger sign in the globus pallidus on T2*-weighted magnetic resonance imaging which reflects the focal accumulation of iron in this area (Zorzi *et al*, 2011).

PANK2-mediated CoA biosynthetic pathway takes place in mitochondria and cytosol. It involves five consecutive enzymatic steps highly conserved in animal evolution, starting from pantothenate (vitamin B5), ATP, and cysteine (Leonardi *et al*, 2005; Srinivasan *et al*, 2015). The recent identification of Coenzyme A synthase (COASY, the enzyme catalyzing the last two steps of CoA biosynthesis) as causative of a subtype of NBIA (Dusi *et al*, 2014) strongly reinforces the essential role of CoA in the correct functioning of the neural cells. In fact, CoA is a key molecule involved in more than 100 metabolic processes, among which CoA derivatives are crucial substrates for ATP generation via the tricarboxylic acid cycle, fatty acid metabolism, cholesterol and ketone body biosynthesis, and histone and non-histone protein acetylation (Siudeja *et al*, 2011; Akram, 2014). Although these processes are vital for any cell type, it remains unexplained why the disease affects primarily the central nervous system.

Animal models for PKAN, despite being informative for studying pathological mechanisms, share only a few neuropathological signs—with milder severity—compared to those associated with the human disorder, limiting their impact for predicting novel therapeutics (Kuo *et al*, 2005; Rana *et al*, 2010; Brunetti *et al*, 2012; Garcia *et al*, 2012). In fact, although presenting some levels of neurodegeneration, these models lack any evidence of brain iron

1   Proteomics of Iron Metabolism Unit, Division of Neuroscience, San Raffaele Scientific Institute, Milan, Italy
2   Stem Cells and Neurogenesis Unit, Division of Neuroscience, San Raffaele Scientific Institute, Milan, Italy
3   Neuroimmunology Unit, Division of Neuroscience, San Raffaele Scientific Institute, Milan, Italy
4   Molecular Neurogenetics Unit, Foundation IRCCS-Neurological Institute "Carlo Besta," Milan, Italy
5   Genomic Unit for the Diagnosis of Human Pathologies, Division of Genetics and Cell Biology, San Raffaele Scientific Institute, Milan, Italy
6   Vita-Salute San Raffaele University, Milan, Italy
7   Institute of Neuroscience, National Research Council, Milan, Italy
    *Corresponding author. Tel: +39 02 26434755; Fax: +39 02 26434844; E-mail: levi.sonia@hsr.it
    †The authors share the first authorship
    ‡The authors share senior authorship

mishandling, preventing any insight on the causative link between PANK2 deficiency and brain iron deposition (Levi & Finazzi, 2014). In vitro studies of PKAN patients' fibroblasts have been instrumental in revealing some defects in mitochondrial activity and iron metabolism associated with PANK2 deficiency, but their specific contributions to the pathological neurodegenerative processes cannot be ascertained in these cells (Campanella et al, 2012; Santambrogio et al, 2015).

Experiments in cell cultures have revealed that pantetheine, in addition to vitamin B5, can also be phosphorylated, and its product, 4′-phosphopantetheine, can function as a precursor for CoA (Srinivasan et al, 2015). Interestingly, food supplemented with pantetheine was shown to partially revert the neuronal defects in mutant flies, fish, and mice (Rana et al, 2010; Brunetti et al, 2014; Zizioli et al, 2015). However, pantetheine-mediated rescuing effects were particularly limited in the mouse model, since this molecule is highly unstable in serum and is rapidly converted into vitamin B5 and cysteamine by pantetheinases (Brunetti et al, 2014). Recently, an alternative mechanism of CoA delivery to the cells has been described, consisting in extracellular CoA conversion into 4′-phosphopantetheine, which in turn passively crosses membranes and is converted back into CoA by COASY (Srinivasan et al, 2015). Notably, neurodegeneration in dPANK-deficient mutant flies and fish was rescued by CoA administration by raising the levels of total intracellular CoA (Srinivasan et al, 2015; Zizioli et al, 2015). However, it remains unknown whether exogenous CoA administration can be efficacious in more complex animal models or in human cells.

Considering the aforementioned limitations in the existing cellular and animal models, we took advantage of the human induced pluripotent stem cell (hiPSC) reprogramming technology to establish cultures of faithful human neuronal cells starting from patients' fibroblasts (Marchetto et al, 2011; Tiscornia et al, 2011; Peitz et al, 2013; Amamoto & Arlotta, 2014; Broccoli et al, 2015).

Here, we demonstrate that PKAN hiPSC-derived neurons exhibited severe functional impairments such as alteration of the oxidative status and mitochondrial dysfunctions, including impaired energy production, iron–sulfur cluster (ISC) and heme biosynthesis, with consequent cellular iron imbalance. Strikingly, supplementation of CoA in the neuronal growth medium was sufficient to restore the majority of these functionally defective phenotypes.

## Results

### Generation and characterization of PKAN and normal subjects hiPSC-derived neurons

To obtain a human PKAN neuronal model, we established multiple lines of transgene-free hiPSCs by reprogramming fibroblasts from three patients and three neonatal normal subjects (here referred to as controls) by Sendai virus-mediated expression of the four Yamanaka's factors (see Materials and Methods). One patient carried the c.569insA mutation, causing the premature stop codon p.Y190X, while two siblings were carrying the same mutation c.1259delG causing a frameshift p.F419fsX472 (here referred to as p.F419fsX472a and b). These mutations lead to the complete lack of PANK2 protein in fibroblasts (Santambrogio et al, 2015).

Independent hiPSC clones for each individual were generated and fully characterized. In particular, we selected one clone from each healthy subject (controls 1, 2, 3), three from patients p.F419fsX472a (#3, 5, 8) and b (#3, 5, 11), and one from patient p.Y190X (#1). All the different clones were subjected to the same reported analysis, of which an example is provided in each figure.

The expression of master regulators of pluripotent stem cells and associated markers assessed by RT–PCR (FGF4, GDF3, REX1, TERT, KLF4, SOX2, c-Myc, DPPA-2, DPPA-4, OCT4, and TDGF), as well as immunofluorescence analysis (OCT4, SSEA-1, NANOG, TRA1-60 and SOX2), confirmed the stem cell pluripotency state of the hiPSC lines (Fig EV1A and B). Additionally, in all clones NANOG expression was detected at high levels as compared to controls' fibroblasts evaluated by qRT–PCR (Fig EV1C).

Furthermore, pluripotency of hiPSC lines was functionally demonstrated by their effective differentiation into cells of the three different germ layers expressing endodermal, mesodermal, and ectodermal markers (Fig EV1D). The presence of the original PANK2 mutations was verified by direct sequence analysis of each hiPSC clone (Fig EV2A). All the selected hiPSCs were regularly assessed for the maintenance of correct karyotype content during cell expansion in vitro (Fig EV2B).

We then differentiated control and PKAN hiPSCs into a pure and stable population of self-renewable neuronal precursor cells (NPCs). To this end, hiPSCs were differentiated into embryoid bodies (EBs) (Fig EV3A) in the presence of strong inhibitors of the SMAD signaling until the emergence of neural-like rosettes composed of radially organized Nestin$^+$ neural progenitors expressing the forebrain-specific genes Pax6, FoxG1, Tbr2, and Ctip2 with equal intensity in both control and PKAN cell lines (Fig EV3B). On day 21, neural rosettes were isolated, disaggregated, and transferred to N2/B27-based medium supplemented with the growth factor FGF2 (Marchetto et al, 2010). In these conditions, FoxG1+ NPCs gave rise to long-term expandable populations of highly proliferative cells. Stable NPC cultures were established with equal efficiency from all controls and PKAN hiPSCs (Fig EV3C) and were competent in differentiating into neurons and astrocytes with comparable efficiency (Fig EV4A).

Effective in vitro modeling of disease relies on the generation of human neurons with substantial functional activity. To this end, we opted to overexpress the neurogenin-2 (Ngn2) neurogenic factor, which was shown to dramatically accelerate neuronal maturation in vitro and generate a large amount of enriched glutamatergic neurons (Zhang et al, 2013; Ho et al, 2016; Yi et al, 2016). Thus, after verifying the absence of PANK2 in NPCs (Fig 1A), they were transduced with a lentivirus co-expressing both Ngn2 and the puromycin resistance gene to select only for the transgene-expressing cells. Neuronal differentiation was promoted in a B27 serum-free medium supplemented with BDNF. Two weeks after Ngn2 expression, control and PKAN neurons appeared to have developed a complex morphology, organized in a dense network and expressing crucial neuronal markers like Tuj1, MAP2, and NeuN (Fig 1B). Immunofluorescence analysis also revealed that hiPSC-derived neurons expressed the voltage-gated Na$^+$ channels (panNav) and that the majority of them expressed vesicular glutamate transporter 1 (VGlut1) (Figs 1B and EV4B). These data suggested that the majority of the neurons generated by Ngn2 expression was glutamatergic (Fig EV4B), as previously shown by Zhang et al (2013)

and recently confirmed in other studies (Ho *et al*, 2016; Yi *et al*, 2016). The complete absence of the 58kDa band, corresponding to the PANK2 protein, was shown through immunoblotting in the three PKAN patients' derived neurons, which completely lack PANK2 expression due to stop codon and frameshift mutations (Fig 1C). PANK2 was also detected at high levels in PKAN mutant neurons transduced with a functional copy of *PANK2* by lentiviral transduction (PANK2-LV) before the induction of differentiation (Fig 1C). Additionally, the anti-human PANK2 antibody recognizes an unspecific band of lower molecular weight (asterisk in Fig 1C). Morphological inspection did not reveal any difference in either total dendritic length or branching complexity when comparing control and PKAN neurons (Fig 1D).

In order to evaluate neuronal firing properties, we performed electrophysiological recordings at 4–20 weeks from initial differentiation. To obtain comparable growth conditions and to minimize interferences, control and PKAN neurons were co-cultured in the same culture dish and were distinguished by expression of either the GFP (control) or tdTomato (PKAN) fluorescent proteins (Fig 1E). Mouse cortical neurons obtained from E18 embryos were added to improve electrophysiological activity (Verpelli *et al*, 2013). Individual neuronal cells were first recorded in current-clamp mode to detect intrinsic properties and action potential firing activity. The majority of control cells (63 ± 8%, 20 out of 32) responded to injection of suprathreshold current steps (10–100 pA, 1 s) with trains of overshooting action potentials at variable frequencies (5–15 Hz). Conversely, the majority of PKAN neurons (61 ± 12%, 14 out of 23) responded to similar current injection protocols with an anomalous firing activity consisting of brief series of spikes (often no more than 2–3 action potentials with strongly decremental amplitudes) followed by a plateau potential superimposed with irregular oscillations (Fig 1F). The average maximal firing rate was 13 ± 1 Hz in 14 control and 3 ± 2 Hz in 13 PKAN human neurons, respectively ($P = 0.0004$, unpaired *t*-test), while the mean resting membrane potential was −48 ± 3 mV in control vs. −37 ± 2 mV in PKAN human neurons ($P = 0.01$, unpaired *t*-test). Conversely, the input resistance was not significantly different in the two groups (ctrl: 507 ± 78 MΩ, PKAN human neurons: 677 ± 151 MΩ; $P = 0.14$, unpaired *t*-test). In addition, peak amplitudes of voltage-dependent sodium currents ($I_{Na}$) were significantly larger in control cells than

in PKAN neurons (1970 ± 464 pA vs. 849 ± 207 pA, respectively, $P = 0.04$, Mann–Whitney rank sum test; Fig 1F). These data suggest that human PKAN neurons displayed aberrant electrophysiological properties compared to neurons from normal donors (Fig 1G). Specifically, PKAN neuronal cells showed significantly reduced peak $Na^+$ currents and were unable to respond to current stimulation with appropriate trains of repetitive spikes.

## PANK2 deficiency leads to mitochondrial dysfunction in PKAN human neurons

To address whether the affected functionality of PKAN neurons was associated with mitochondrial dysfunction, we first evaluated the integrity of mitochondrial membrane potential using the mitochondria-specific fluorescent probe tetramethylrhodamine methyl ester (TMRM) (Cozzi *et al*, 2013). At 3 weeks from differentiation, the fluorescence associated to the neuronal cells, as recognized by neuronal specific anti-NCAM staining, was sampled by the IN Cell Analyzer system across the entire neuronal culture—thus avoiding limitations with a manual inspection (Fig 2A). The mitochondrial uncoupler carbonyl cyanide 4-(trifluoromethoxy)phenylhydrazone (FCCP) completely abolished the mitochondrial-specific fluorescence of TMRM probe (Fig 2A). Results were plotted relative to the mean of control fluorescence intensity, showing that PKAN neurons exhibited a statistically significant reduction (about 20% lower) in TMRM incorporation with respect to control neurons (Fig 2A). Ultrastructural analysis revealed evident morphological alterations of mitochondria in PKAN neurons, which appear aberrant, enlarged, and swollen with damaged cristae—often tightly packed against the outer membrane with vacuolization of the matrix (Fig 2B). By measuring the longer diameter perpendicular to the longitudinal axis of mitochondria ($n = 200$), we revealed a difference in the distribution of diameter length, which suggests that a higher proportion of altered mitochondria is present in PKAN compared to control neurons (Fig 2B plot). Remarkably, the abnormal mitochondrial phenotype and size observed in PKAN neurons were prevented by reintroducing a functional copy of *PANK2* before differentiation (Fig 2B).

Next, we investigated respiratory activity as a critical parameter of mitochondrial function. Respiration was quantified by microscale

---

**Figure 1.   Development and characterization of hiPSC-derived neurons from controls and PKAN patients.**

A   Representative IF image of NPCs stained for Nestin, FoxG1, and Pank2.

B   NPCs differentiated into neurons by overexpressing Ngn2 (one representative experiment is shown). Two weeks after the infection differentiated NPC were positive for neuronal markers βIII tubulin (Tuj1), Map2, NeuN and human nuclei (hNu) and synaptic markers, the voltage-gated $Na^+$ channels (PanNav), and the vesicular glutamate transporter 1 (VGlut1).

C   Western blot of soluble cell homogenates from human neurons probed with PANK2 and β-actin antibodies (arrows). Asterisk indicates nonspecific band. Data are representative of three independent experiments.

D   Plots showing the total dendritic length and branching points. Data presented as mean + SEM from at least three independent experiments. A total of 38 neurons were counted for each sample. Statistics were determined by the *t*-test and resulted not significant.

E   Representative example of co-culture containing control (green), PKAN (red) human neurons, and E18 cortical mouse neurons. Control and PKAN NPCs were infected with GFP-LV and tdT-LV expressing vectors, respectively, and differentiated for 8 weeks.

F   Examples of electrophysiological properties of human neurons: control individual (top); PKAN patient (bottom). Traces on the left represent trains of action potentials induced by injection of a suprathreshold current step through the patch electrode in current-clamp mode. Middle traces show $Na^+$ and $K^+$ currents (down- and upward-deflecting from baseline, respectively) in response to a 60 mV step from a holding voltage of −70 mV in voltage-clamp mode. Insets on the right display enlarged portions of the traces to magnify fast $Na^+$ currents.

G   Histogram with percentages of recorded cells showing repetitive firing. Data presented as mean + SEM, Controls $n = 32$, PKAN patients $n = 23$ (*z*-test).

Data information: (A, B, and E) Scale bars 20 μm.

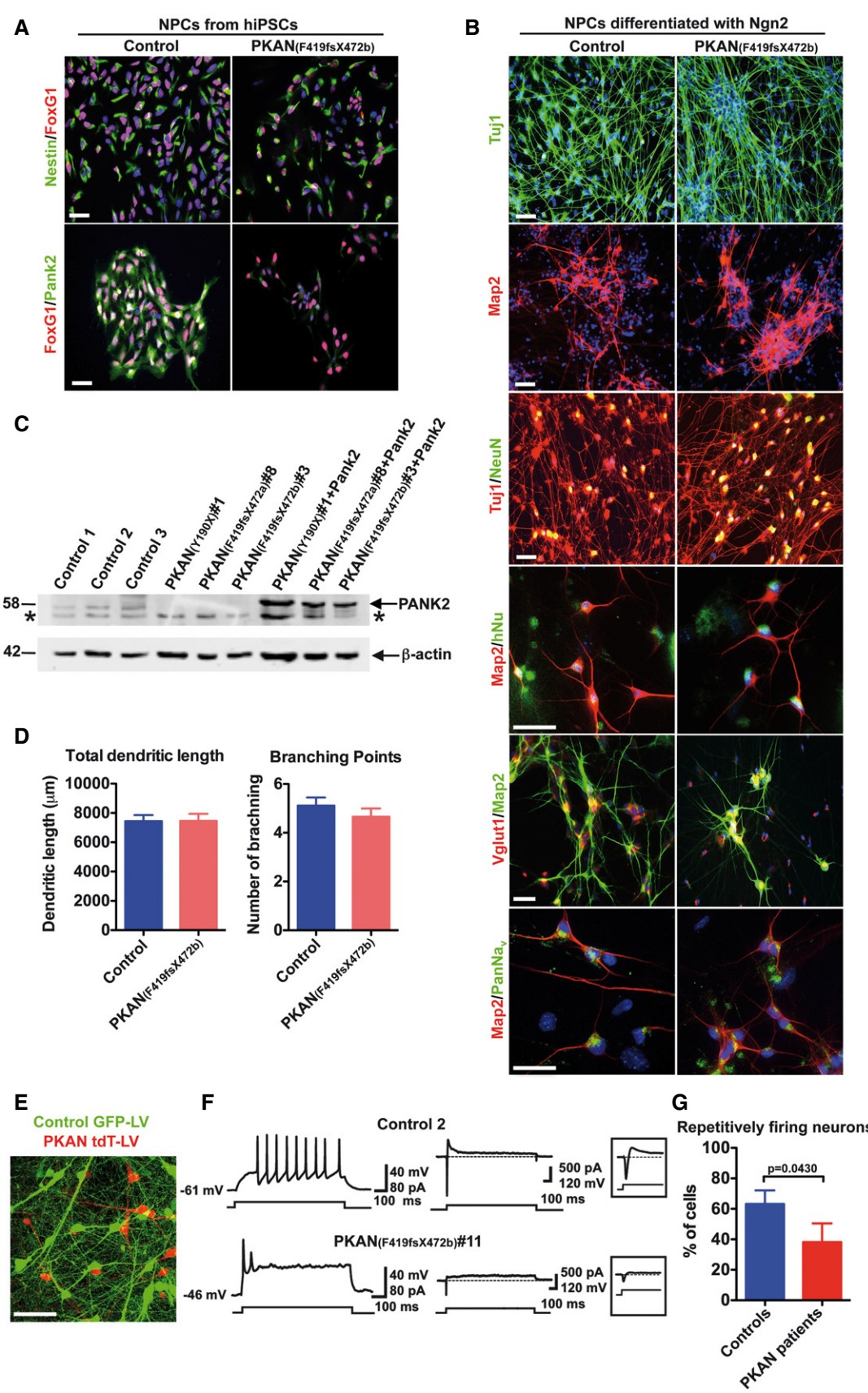

**Figure 1.**

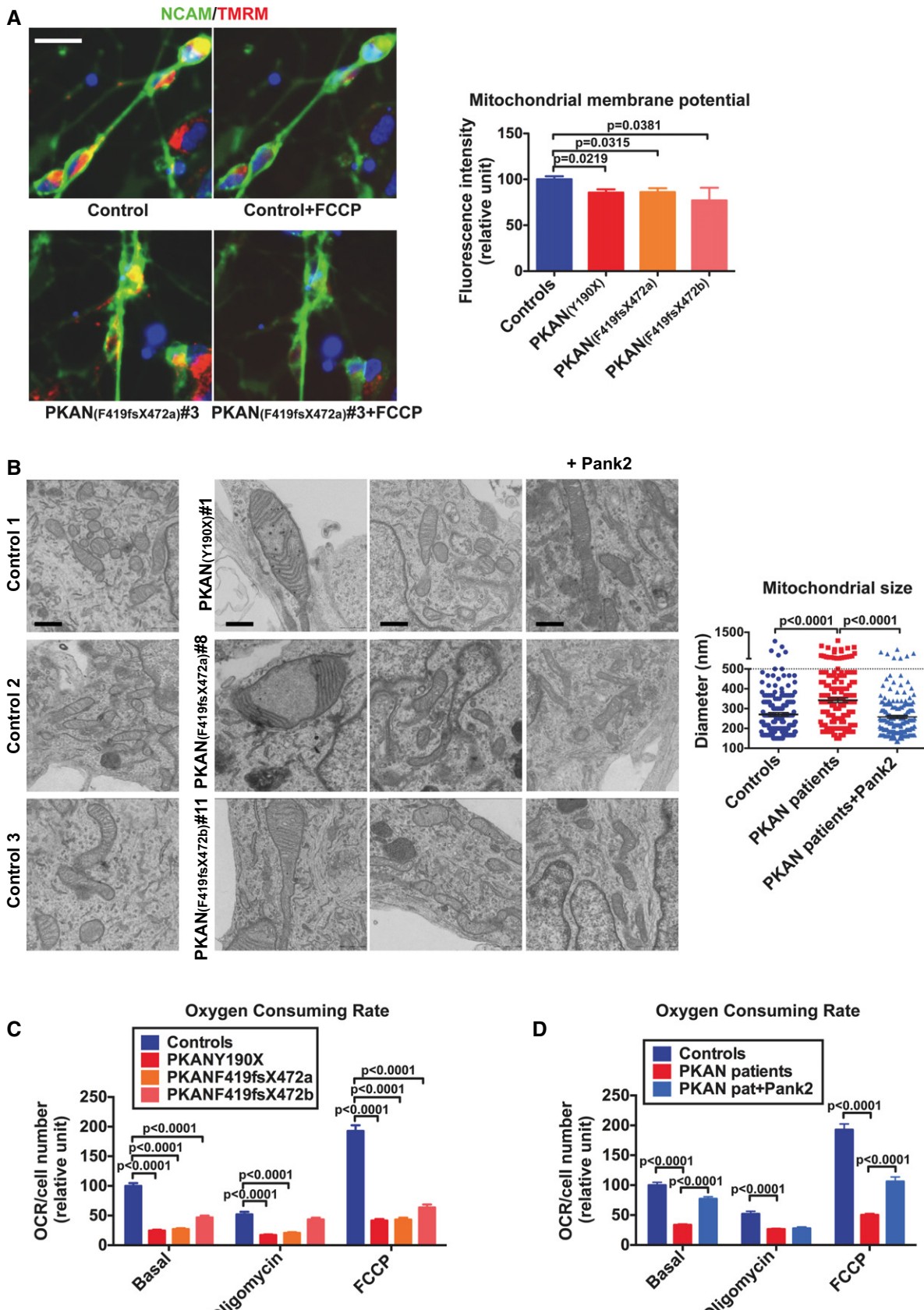

**Figure 2.**

◄

**Figure 2. Mitochondrial membrane potential and morphology were affected in PKAN human neurons.**

A Representative images of human neurons cells stained with the mitochondrial membrane potential-sensitive fluorescent probe TMRM, the neuronal-specific anti-NCAM antibody, and the nuclear-staining Hoechst. Left panel, basal conditions. Right panel, after addition of the mitochondrial uncoupler FCCP. Scale bar 20 μm. Plot showing the quantification of TMRM fluorescence signal from NCAM$^+$ neurons. Data presented as means + SEM of three independent experiments (unpaired, two-tailed $t$-test).

B Representative images of ultrastructural analysis of fixed neurons examined with electron microscope. PANK2 panel represents PKAN neuronal cells overexpressing PANK2 protein. Scale bar 500 nm. Mitochondrial size was measured at level of the larger diameter along the perpendicular axis for all the mitochondria in > 30 fields (200 mitochondria in total) for each sample (one-way ANOVA).

C, D (C) OCR measurements of controls and each PKAN patients analyzed individually and (D) data obtained by individual analysis and plotted as grouped controls, PKAN patients, and PKAN patients overexpressing PANK2. The plots show OCR normalization to cell number. OCR was measured in basal conditions, and after oligomycin and FCCP addition. Bars indicate means + SEM of three independent experiments (two-way ANOVA).

oxygraphy, allowing the real-time measurement of the global oxygen consumption rate (OCR) (Fig 2C). Oligomycin and FCCP treatments were also performed in order to measure ATPase inhibition and the uncoupled stimulated respiration. The values obtained for the three PKAN patients were significantly lower than those obtained in control neurons for each of the respiratory conditions, except for the oligomycin treatment of the PKAN patient with the mutation F419fsX472b. This reduced respiratory ability is in agreement with the presence of mitochondrial dysfunctions. As expected, PANK2 re-expression, confirmed by immunoblotting (Fig 1C), was sufficient to revert OCR levels in PKAN neurons (Fig 2D).

### PANK2 deficiency alters the oxidative status of PKAN neurons

One of the downstream effects of impaired respiration is the increase of radical oxygen species (ROS). Thus, we monitored ROS levels in basal conditions using the fluorescent ROS-sensitive dichlorofluorescein (DCF) on 3-week differentiated NCAM-positive neurons (Fig 3A). Interestingly, ROS levels were strongly enhanced in the PKAN compared to control neurons (Fig 3A). As expected, PANK2 re-expression (Fig 1C) was sufficient to reduce ROS levels in PKAN neurons (Fig 3A), confirming that this altered oxidative status is directly related to PANK2 deficiency. In addition, we measured the reduced form of glutathione using the ThiolTracker Violet probe in Tuj1-positive neurons (Fig 3B). In line with heightened ROS levels, significantly lower levels of reduced glutathione were detected in PKAN compared to control neurons (Fig 3B).

### PANK2 deficiency altered mitochondrial iron-dependent biosynthetic pathway and cytosolic iron homeostasis

In mitochondria, iron is converted into its biological active form through two iron-dependent biosynthetic pathways: ISC and heme. To verify whether PANK2 deficiency leads to impairment of these mitochondrial pathways, we investigated the activity of two ISC-containing enzymes and the heme content in hiPSC-derived neurons. In-gel activities of mitochondrial and cytosolic aconitases (mAco and cAco) were measured in 3-week differentiated neurons (Fig 4A, upper panel). A significant reduction in activity of both aconitases was detected in PKAN compared to control neurons (Fig 4A, lower panel). This decrease was not due to reduced protein levels, since comparable amounts of mAco and cAco were revealed by Western blot analysis (Fig 4B).

Heme quantification was performed only on NPCs (not on neurons) due to the low sensitivity of the method, which requires a substantial amount of cells. The spectroscopic quantification

revealed a significant reduction of heme in PKAN compared to control NPCs (Fig 4C). Concomitant decrease in both ISC and heme content might represent a signal for the cell to enhance iron incorporation through the activation of the mRNA-binding activity of the ISC-deprived form of cAco (apo-cAco), which controls the translation of mRNAs of several iron-related proteins (IRP1/IRE machinery) (Muckenthaler *et al*, 2008). We tested this hypothesis by measuring the level of two iron proteins responsible for either cellular iron uptake (transferrin receptor1, TfR1) or iron storage (ferritin, FtH). As expected by the elevated amount of the apo-cAco form, the level of TfR1 was increased (~1.8-fold) while ferritin was reduced (~3-fold) in PKAN compared to control neurons (Fig 4D). These results provide evidence that iron metabolism is impaired in PKAN neurons, which exhibit a manifested cellular iron-deficient phenotype.

### Exogenous CoA can rescue the deficits in PKAN neurons

The evidence that cells can use external CoA as a source for internal CoA biosynthesis (Srinivasan *et al*, 2015) prompted us to verify the effect of its addition to the neuronal culture medium. In a first instance, we monitored the vitality of PKAN and control neuronal co-cultures over time after differentiation. Analysis was performed by counting the number of either tdTomato$^+$-PKAN or GFP$^+$-control neurons as shown in Fig 5A. No difference between the two neuronal populations was detected after 1 day of differentiation. At day 120, about 50% of PKAN neurons were lost compared to day 1 with significant difference respect to the control neurons that decreased only by 12% (Fig 5A). At day 150, the number of PKAN neurons dropped to about 20% while control neurons were only slightly reduced (Fig 5A). This might well be caused by the severe impairment of PKAN neurons in acquiring functional properties, which promote neuronal survival. In contrast, neuronal growth medium supplemented every two days with 25 μM CoA (starting at day 120) strongly reduced PKAN neuronal loss at day 150, promoting a threefold increase in surviving neurons (Fig 5A). In addition, patch-clamp experiments revealed a significant recovery of functional properties in PKAN neurons (Fig 5B). The rate of PKAN neurons with mature firing increased from 33 ± 13% (8 out of 21 cells) in untreated conditions to 79 ± 15% (11 out of 14 cells, $P = 0.018$, $z$-test) after CoA treatment. Conversely, the rate of control neurons with mature functionality was not different in CoA vs. untreated conditions (7 out of 12 cells, 58% and 14 out of 20 cells, 70%, respectively, $P = 0.25$, $z$-test; Fig 5B).

Next, we examined whether the addition of CoA from day 1 onwards during differentiation could restrain the heightened

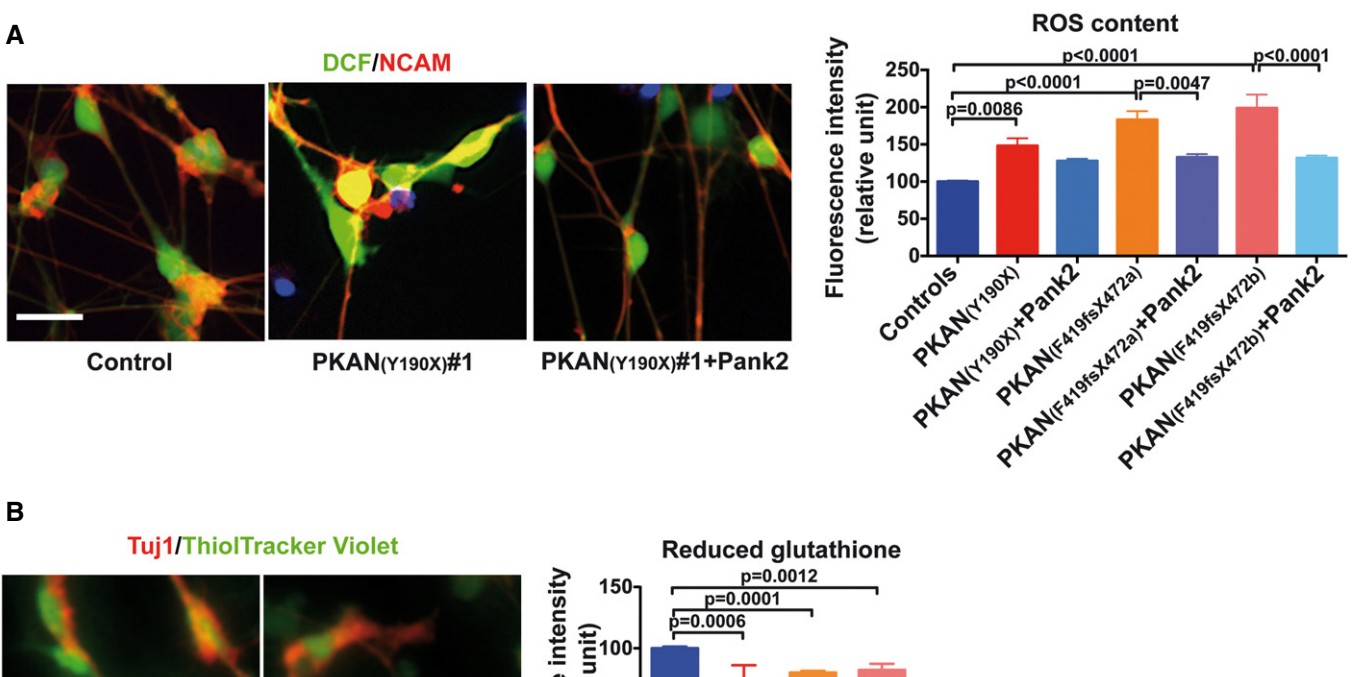

**Figure 3. PKAN human neurons show altered oxidative status.**

A  Representative images of neurons stained with the neuron-specific anti-NCAM antibody, ROS-sensitive fluorescent probe DCF, and the nuclear dye Hoechst. Scale bar 20 μm. Plots of the DCF fluorescence signal from NCAM-positive control and PKAN human neurons, infected or not with Ngn2-PANK2-LV. Data presented as means + SEM of at least three independent experiments (one-way ANOVA).

B  Representative images of neurons stained with ThiolTracker Violet and the anti-Tuj1. Scale bar 20 μm. ThiolTracker Violet fluorescence signal from Tuj1-positive neurons was quantified and shown in the plots. Data presented as means + SEM of at least three independent experiments (unpaired, two-tailed *t*-test).

oxidative status in PKAN neurons. Indeed, in the presence of CoA, the ROS levels were comparable between PKAN and control neurons, indicating that CoA could effectively restrain the disease-associated ROS overproduction (Fig 5C). Notably, the presence of CoA restored the mitochondria respiratory activity to control levels for each of the respiratory conditions in the three patients' derived PKAN neurons (Fig 5D). To check the beneficial effect of CoA treatment on iron-dependent mitochondrial biosynthesis, we quantified the amount of heme, which was recovered in PKAN NPCs (Fig 5E).

## Discussion

Pantothenate kinase-associated neurodegeneration is a devastating infantile disorder for which only symptomatic treatments are currently available. The cascade of pathophysiological events caused by defective CoA biosynthesis as well as the association between mitochondrial dysfunction and brain iron accumulation are

far from being clear. One prevailing hypothesis proposes that the imbalance of CoA pool could impair lipid homeostasis, resulting in membrane dysfunction and mitochondrial alteration including energy deficiency and impairment of oxidative status and iron metabolism. Indeed, recent data obtained by microarray and whole-transcriptome gene expression assays indicated an interconnection between NBIA genes and iron-related genes. These genes might be implicated in synapse and lipid metabolism-related pathways (Bettencourt *et al*, 2015; Heidari *et al*, 2016).

Our previous results obtained on PKAN fibroblasts and neuronal cells generated by direct conversion of fibroblasts suggested that this disease is associated with mitochondria functional alterations and related hyper-oxidative status (Santambrogio *et al*, 2015). However, these two models present significant limitations concerning the opportunity to clarify the relationship between iron dysregulation, neuronal functions, and cell death. This is essentially due to the cell-specific iron metabolism requirement in fibroblasts, which is significantly different from that in neurons, and to the poor efficiency of direct conversion of fibroblasts into neurons, which

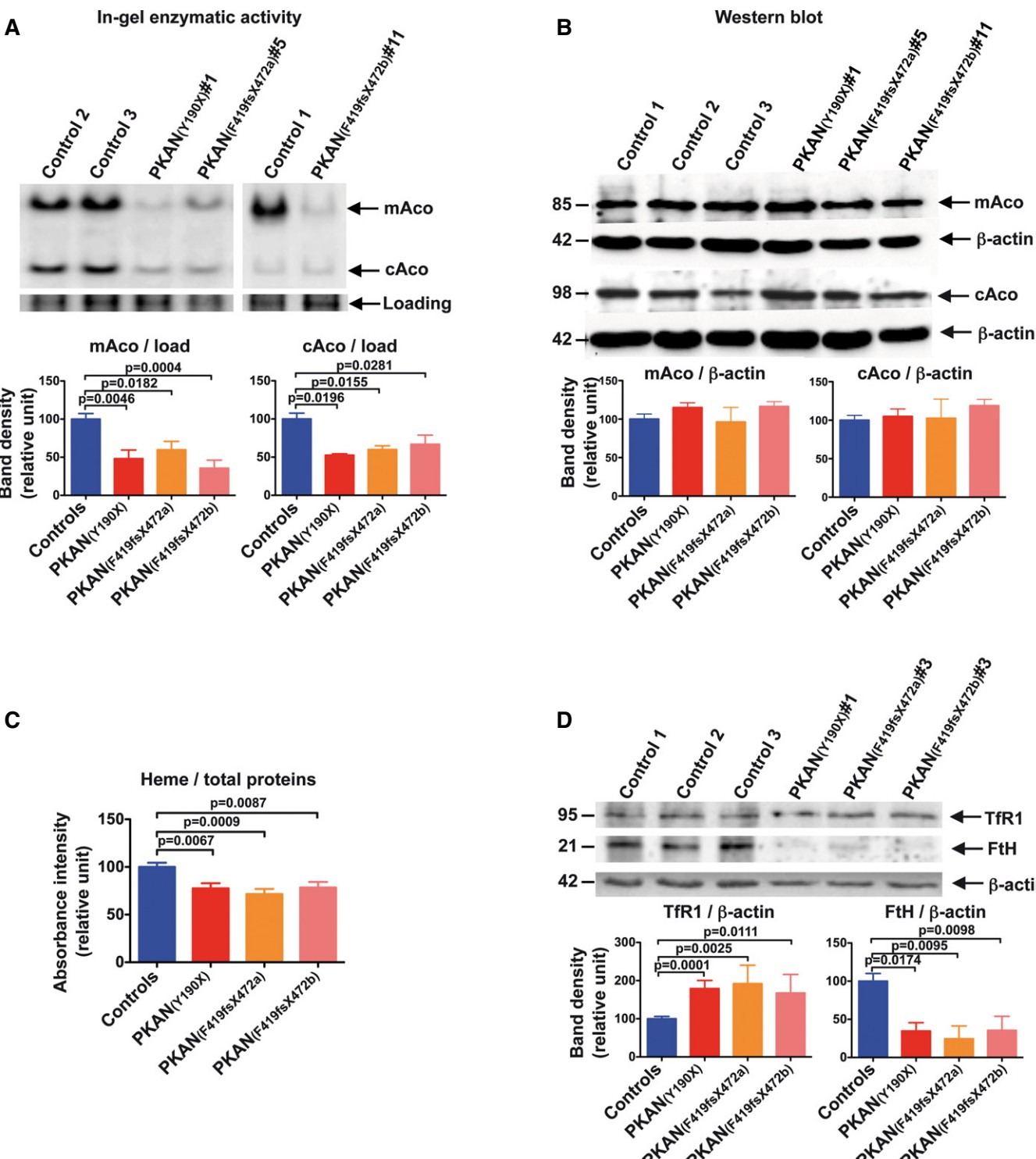

**Figure 4.  PKAN human neurons show impaired iron-dependent mitochondrial biosynthesis.**

A    Upper panel: representative images of in-gel enzymatic activity of mitochondrial and cytosolic aconitases (mAco and cAco, respectively). The lower part of the gel was cut, stained with Coomassie blue, and a protein band was used as loading control (Loading). Lower panel: quantification of mAco and cAco enzymatic activity by densitometry.

B    Upper panel: Western blot analysis of mitochondrial and cytosolic aconitases. Lower panel: quantification of mAco and cAco by densitometry.

C    Heme quantification by absorbance at 400 nm of the soluble cell lysates.

D    Upper panel: Western blot analysis of transferrin receptor (TfR1) and ferritin (FtH). Lower panel: quantification of TfR1 or FtH normalized on actin by densitometry.

Data information: All the data are presented as means + SEM of at least three independent experiments (unpaired, two-tailed $t$-test).

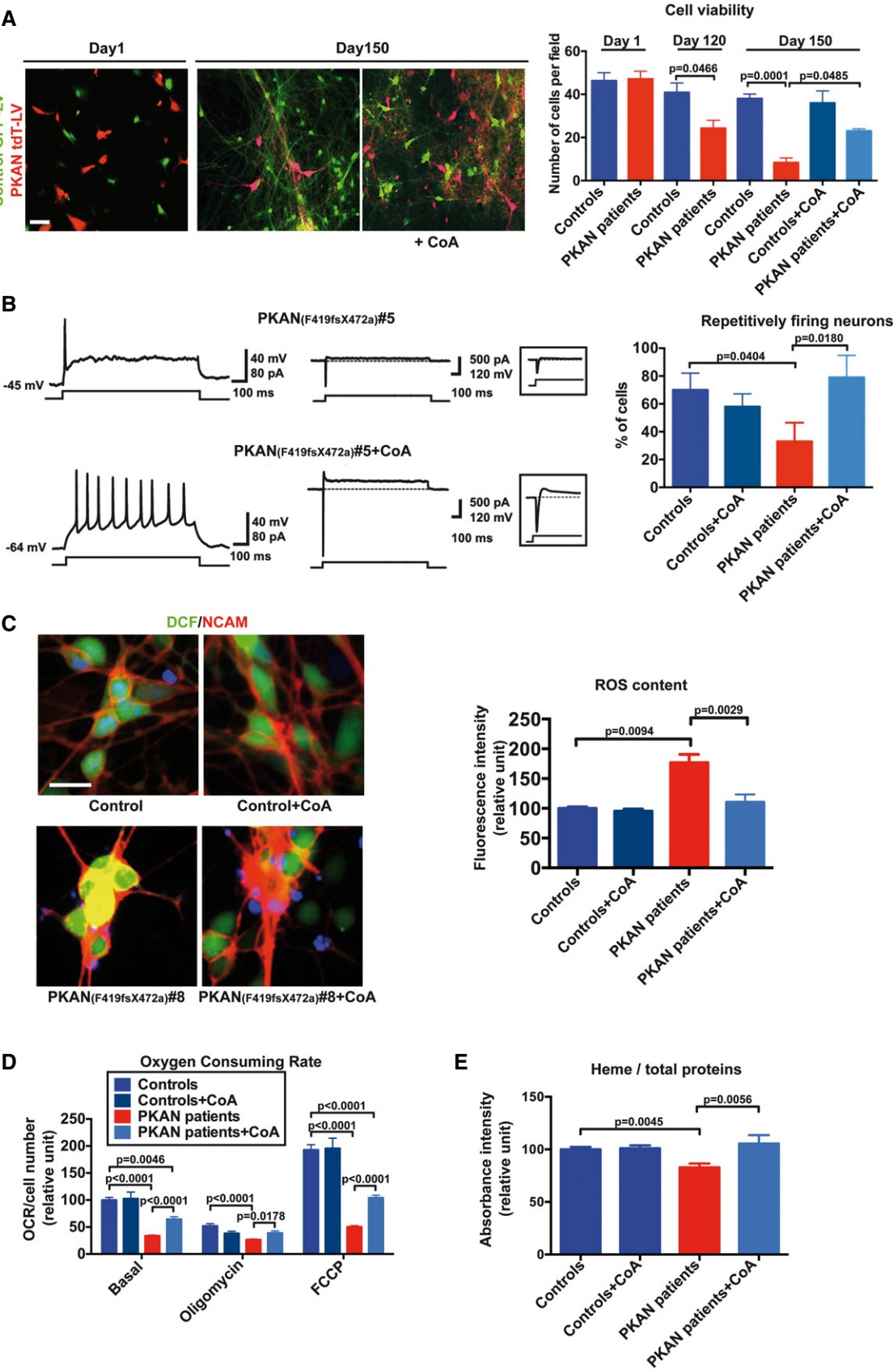

**Figure 5.**

◄

**Figure 5. CoA treatment recovers PKAN human neurons abnormal phenotype and functionality.**

A  Representative images of co-cultures of control and PKAN NPCs infected with GFP-LV and tdT-LV expressing vectors, respectively, at the beginning (Day 1), after 120 (Day 120) and 150 days of differentiation (Day 150). Scale bar 20 μm. Plots show the number of green and red human neurons counted at different time points. All data are presented as means + SEM of at least three independent experiments (one-way ANOVA).

B  Example of electrophysiological properties of cultured PKAN human neurons with or without CoA incubation for 30 days. Repetitive firing activity (left) and relatively large $Na^+$ and $K^+$ currents (right) were restored by CoA. The histogram on the right shows fractions of repetitively firing cells recorded in untreated vs. CoA-treated neurons from control and PKAN patients. Data presented as mean + SEM, Controls $n = 20$, Controls+CoA $n = 12$, PKAN patients $n = 21$, PKAN patients+CoA $n = 14$ (z-test).

C  An example of neurons stained with the ROS-sensitive fluorescent probe DCF and the nuclear dye Hoechst. Anti-NCAM antibody was used to detect neurons. Scale bar 20 μm. Plots of the DCF fluorescence signal from $NCAM^+$ control and PKAN human neurons differentiated or not in the presence of CoA (25 μM) in the medium for 3 weeks. All data are presented as means + SEM of at least three independent experiments (one-way ANOVA).

D  Oxygen consumption rate (OCR) with and without CoA on controls and PKAN patients. Basal and uncoupled (FCCP) respiration increased after CoA supplementation. Data presented as means + SEM of 24 independent replicates for each condition (two-way ANOVA).

E  Heme quantification by absorbance at 400 nm of the soluble NPC cell lysates. All data are presented as means + SEM of at least three independent experiments (one-way ANOVA).

hampers any functional electrophysiological and biochemical investigation (Santambrogio et al, 2015).

Herein, we have established hiPSC-derived neuronal cultures with high efficiency from PANK2-deficient patients, providing an in-depth molecular and biochemical characterization of their metabolic and functional alterations that allowed us to exhaustively define the neuronal phenotype. In addition—and more significantly—we demonstrated, for the first time, the therapeutic effect of exogenous CoA administration in reverting pathological phenotypes in PKAN-derived neurons.

The first result of our work is the successful generation of PKAN hiPSCs and their neuronal derivatives, suggesting that PANK2 deficiency does not affect the neuronal fate commitment and differentiation of these cells, as would be expected considering that the patients have normal brain development.

We also provided evidence that PKAN neurons displayed profound alterations of mitochondrial morphology and energetic capacity, which were probably responsible for the loss of fully functional neurons. Indeed, most PKAN neurons were impaired in their ability to fire repetitive trains of action potentials in response to depolarizing current injection, a defect that may derive either from the inability of neuronal precursors to mature into functional neurons, or from a dissipation of energy-dependent transmembrane ionic gradients, which in normal conditions ensure the appropriate flux of ion currents through membrane channels.

In addition, we attributed the alteration of iron homeostasis to two defective mitochondrial iron-dependent pathways: ISC and heme biosynthesis. ISCs are prosthetic groups of numerous mitochondrial and cytosolic enzymes (respiratory complexes, ferrochelatase, aconitases, lipoate synthase, DNA helicases, and others) (Lill et al, 2014). Thus, their deficiency may affect many biosynthetic pathways and trigger neurodegeneration. Similar neurodegenerative events have been described to occur in Friedreich's ataxia (Pandolfo, 2003), where deficiency of the ISC–iron chaperone, frataxin, causes shortages of the ISC-dependent respiratory complexes that lead to energy deficiency (Lodi et al, 1999; Hick et al, 2013). This deficit is responsible for a decrease in antioxidant capacity and accumulation of iron in mitochondria due to its inefficient utilization. The consequent damage is primarily caused by iron-generated free radicals that may inflict further injury to ISC-proteins (Lu & Cortopassi, 2007). Interestingly, our results highlighted a comparable pathological association between ISC defects and energy deficiency, suggesting that these factors are common

features in triggering neuronal death. Although further analyses are necessary to clarify the link between ISC defect and energy deficiency, we can speculate that the ISC-dependent mAco decreased functionality may also worsen the efficacy of the Krebs cycle, which is impaired in PKAN neurons due to CoA deficiency, decreasing the amount of available NADH and GTP (Pandey et al, 2015). Furthermore, the ineffective mitochondrial iron utilization might be the signal that promotes iron import into the cells and, with time, leads to iron accumulation. This might occur because the cytosolic Aco has variable functions, depending on the presence of ISC cofactor in its structure (Beinert et al, 1996). When ISC is bound to the active site, cAco acquires enzymatic activity and converts citrate to isocitrate. Conversely, its ISC-free apoform (called IRP1) binds the iron responsive sequences (IREs) located on several iron–protein mRNAs, such as the iron-storage ferritin and the iron importer TfR1 (Rouault, 2006). Through this ISC-switching mechanism, the IRP1/IRE machinery senses cytosolic iron status and plays a coordinate control of the proteins involved in iron management to maintain intracellular iron homeostasis (Muckenthaler et al, 2008). Thus, shortage of ISC availability might be the trigger for both energy impairment and iron accumulation. Previous studies (Bettencourt et al, 2015; Heidari et al, 2016) postulated that disturbances in NBIA gene networks could contribute to dysregulation of iron metabolism and, in turn, a progressive increase in brain iron levels aggravating the disruption of these NBIA genes. Nonetheless, we have not detected any frank iron deposition in PKAN neurons yet. Two different reasons might account for this result. On one hand, the period of time where human neurons in culture were observed might not be long enough considering that iron deposition is detectable in patients for only a few or more years after birth. On the other hand, iron accumulation in patients is restricted to pallidal GABA-ergic and, less frequently, dopaminergic neurons of substantia nigra. We preferred to conduct our study on forebrain-specific glutamatergic excitatory neurons since this system enabled the generation of a homogenous population of neural progenitors with a high proliferative index, thus providing large numbers of neurons with robust functional activities for our analysis. Given the pan-neuronal deficits occurring in this disorder, this cellular model is perfectly suited for investigating the pathophysiological roots at the base of this neuropathology, including the initial steps of the iron mishandling phenotype.

Furthermore, we proved that CoA administration to human neurons restores PANK neuron functionality, inhibits neuronal cell

death, prevents the development of harmful ROS, and rescues heme biosynthesis and respiratory activity, which establishes a strong proof of principle for the use of this compound as a therapeutic agent. This is further corroborated by previous *in vivo* data on fly and zebrafish PKAN models (Srinivasan *et al*, 2015; Zizioli *et al*, 2015), which clearly established the efficacy of CoA treatment in preventing or ameliorating the pathologic events.

Overall, these data indicate that this new human neuronal model represents a powerful platform for investigating pathogenic mechanisms of disease and for testing the efficacy of therapeutic compounds. Our observations pave the way for CoA treatment not only for PKAN disease but also for CoA deficiency-related disorders.

# Materials and Methods

## Plasmid constructions

Construction of the TetO-Ngn2-p2a-hPANK2-2HA-t2a-Puromycin viral vector: the TetO-Ngn2-t2a-Puro (kindly provided by T. C. Südhof) was modified in order to insert in the XbaI site p2a peptide in frame with the existing t2a peptide and multiple cloning site (created by oligonucleotide annealing, Table EV1). The human *PANK2* coding sequence followed by a HA tag was PCR amplified from the pCDNA3.1-hPANK2-HA construct (primers in Table EV1), in order to provide it with AgeI and XbaI, respectively, at its 5′ and 3′. They were then used to insert the coding region in frame with the Ngn2 and puromycin cassette in the intermediate described above.

## Fibroblasts culture and hiPSC generation

Neonatal normal male subject fibroblasts were obtained from ATCC (Controls 1 and 2). A normal female hiPSC line was generated from cord blood stem cells and maintained as feeder-free cells in mTeSR1 (Stem Cell Technologies) (Control 3).

Pantothenate kinase-associated neurodegeneration patient fibroblasts were obtained from the Movement Disorders Bio-Bank available at the Neurogenetics Unit of the Neurological Institute "Carlo Besta" (INCB), Milan, Italy. Two of the PKAN patients were siblings (one 2-year-old male and one 4-year-old female at time of skin biopsy) carrying the mutation c.1259delG causing a frameshift p.F419fsX472 (here referred to as PKAN(F419fsX472a) or PKAN (F419fsX472b), respectively) (Campanella *et al*, 2012; Santambrogio *et al*, 2015). A third patient (one female, 12 years old at time of skin biopsy) carries the c.569insA mutation, causing a premature stop codon p.Y190X (Hartig *et al*, 2006). All subjects gave their written consent for the skin biopsy procedure and for the use of the sample material for research purposes. Patients did not present any associated comorbidity. Fibroblasts were cultured in DMEM high glucose (Life Technologies), 10% FBS (Life Technologies), 2 mM L-Glutamine (Sigma-Aldrich), and 1% penicillin/streptomycin (100U Pen, 100 mg/ml Strep). Fibroblasts were reprogrammed into hiPSCs with the CytoTune-iPS 2.0 Sendai reprogramming kit (Life Technologies) according to manufacturer's instructions. Mutations were corroborated by sequence analysis on hiPSC clones. Colonies started to appear 30 days later, and at around day 40, they were selected

according to their morphology and transferred to a new feeder layer with the same culture conditions (DMEM/F12 (Sigma-Aldrich), 20% KnockOut serum replacement KSR (Life Technologies), 1% P/S, 2 mM L-Glutamine, 1% non-essential amino acids MEM NEAA (Life Technologies), 1 mM NaPyr (Sigma-Aldrich), 0.1 mM beta-mercaptoethanol (Life Technologies), and 10 mg/ml FGF2 (Life Technologies)). Subsequently, hiPSC clones were passed into feeder-free conditions on Matrigel hESC-qualified coated plates (Corning), and the percentage of mTeSR-1 medium (Stemcell Technologies) was gradually increased. hiPSCs were passaged every 5–7 days with ReLeSR (Stemcell Technologies) on Matrigel-coated wells. Fibroblasts and iPSCs were periodically tested for mycoplasma contamination by PCR.

## *In vitro* three germ layers differentiation

Feeder-free hiPSCs were treated with Accutase (Sigma-Aldrich) for 5 min, washed with a mTeSR-1 medium and centrifuged. Cells were then resuspended in DMEM-20% FBS-1% P/S and maintained in the same medium for 20 days. The medium was changed every 2 days.

## Karyotype analysis

Metaphase chromosome preparation was obtained from hiPSCs as follows. Cells were grown on a well dish until they reached an 80–90% confluence. Then, colcemid (1:100, Sigma-Aldrich) was added to the cells for 3 h and incubated at 37°C. Cells were then trypsinized, treated for 30 min using standard hypotonic solution, and fixed (3:1, methanol:acetic acid) for 30 min. Chromosomes were spread on a coverslip and stained with Quinacrine (Sigma-Aldrich) mounted in a McIlvaine buffer. Fluorescence was analyzed in a fluorescent microscope. Images were obtained in blind conditions to the examiner.

## qRT–PCR analysis

RNA was extracted using TRIzol reagent (Sigma-Aldrich) and then retrotranscribed using iScript Super Mix (Bio-Rad). In quantitative real-time PCR, Titan HotTaq EvaGreen qPCR mix (BioAtlas) was used and expression levels were normalized with respect to β-actin expression (primers used are in Table EV2).

## Generation of human neuronal precursors cells and neurons

To obtain NPCs, EBs were formed by dissociation of hiPSC colonies with passaging solution (Miltenyi Biotec) and plating onto low-adherence dishes in mTeSR medium for 10 days, supplemented with Noggin (0.5 mg/ml, R&D System), SB431542 (5 mM, Sigma-Aldrich), N2 ((1:200), Life Technologies), and 1% P/S. In order to obtain rosettes, EBs were plated onto Matrigel growth factor reduced (Corning)-coated dishes in DMEM/F12 (Sigma-Aldrich) plus N2 (1:200); 1% NEAA (Life Technologies) and 1% P/S. After 10 days, rosettes were dissociated with Accutase and plated again onto Matrigel-coated dishes containing NPC medium (DMEM/F12; N2(1:200); B27(1:100, Life Technologies); 1% P/S and FGF2 (20 ng/ml)). Homogeneous populations of NPCs were achieved after 3–5 passages with Accutase in the same conditions.

Neurons were obtained as previously described (Zhang *et al*, 2013) with few modifications. NPCs were transduced with a lentivirus expressing Ngn2 cDNA under the control of a tetracycline-responsive promoter and an LV expressing rtTA. Lentivirus was produced as previously described (Indrigo *et al*, 2010). NPCs were seeded on Matrigel-coated wells and differentiated in medium containing Neurobasal (Life Technologies), BDNF (10 ng/ml, Peprotech), NT-3 (10 ng/ml, Peprotech), B27, P/S, and doxycycline (2 μg/ml, Sigma).

## Coenzyme A treatment

A final concentration of 25 μM Coenzyme A (CoA; Sigma-Aldrich, C4780) was added to treat the cells as previously described (Srinivasan *et al*, 2015). CoA was resuspended in a Neurobasal medium (Life Technologies) and stored at −20°C. Fresh medium containing CoA (25 μM) was added every 2 days to the cells undergoing treatment.

## Immunoblotting

$1 \times 10^5$ cells were seeded on 6 well Matrigel-coated plates and differentiated to neurons. Soluble cellular extracts for immunoblotting were obtained by lysing cells in 20 mM Tris–HCl, pH 7.4, 1% Triton X-100, and a protease inhibitor cocktail (Roche) followed by centrifugation at 16,000 *g* for 10 min. Twenty-five micrograms of total proteins was separated by sodium dodecyl sulfate–12% polyacrylamide gel electrophoresis (SDS–PAGE), and immunoblotting was performed using specific antibodies followed by peroxidase-labeled secondary antibodies (Sigma-Aldrich). The signal was then revealed using the ECL-chemiluminescence kit (GE Healthcare) and detected with ChemiDoc-It Imager (UVP). Total protein contents were measured using the BCA protein assay calibrated with bovine serum albumin (Thermo Fisher Scientific). Antibodies used are listed in Table EV3.

## Immunofluorescence

$8 \times 10^4$ cells were seeded on Matrigel-coated covers and differentiated to human neurons. Cells were fixed in 4% paraformaldehyde and processed as previously described (Cozzi *et al*, 2013).

## Determination of aconitase activity

Aconitase activity was in-gel assayed as described in Tong & Rouault, 2006. The patient and control neurons were grown in a differentiation medium, harvested, washed in PBS, and lysed in a 20 mM Tris–HCl buffer (pH 7.4, 1% Triton X-100, protease inhibitor cocktail, 2 mM citrate, 0.6 mM MnCl$_2$, and 40 mM KCl). Soluble extracts (30 μg) in 25 mM Tris–HCl, pH 8.0, 10% glycerol, bromophenol blue were loaded on PAGE gels containing 8% acrylamide, 132 mM Tris base, 132 mM borate, and 3.6 mM citrate in the separating gel; and 4% acrylamide, 67 mM Tris base, 67 mM borate, 3.6 mM citrate in the stacking gel. The run was performed at 180 V for 2.5 h at 4°C. Aconitase activity was determined in the dark at 37°C by incubating the gel in 100 mM Tris–HCl, pH 8.0, 1 mM NADP, 2.5 mM cis-aconitic acid, 5 mM MgCl$_2$, 1.2 mM MTT, 0.3 mM phenazine methosulfate, and 5 U/ml isocitrate

dehydrogenase. The quantification of the signal was performed using the NIH image software, ImageJ.

## Determination of mitochondrial membrane potential

Human neurons were incubated with Alexa Fluor 488 mouse anti-human CD 56 (anti-NCAM; BD Biosciences) for 1 h, with 20 nM of TMRM (Molecular Probes) for 15 min, and with 2 μg/ml of Hoechst 33342 for 2 min. All of these incubations were performed at 37°C. The cells were washed and randomly analyzed by IN Cell Analyzer 1000 system (GE Healthcare). The fluorescence of TMRM from NCAM-positive cells was collected to compare the relative mitochondrial membrane potential. A minimum of 100 neurons for each patient and control was analyzed in at least three independent experiments for each sample.

## Determination of heme content

Heme content was measured in NPCs from patients and controls as previously described (Santambrogio *et al*, 2011). Briefly, the cells were washed with a phosphate-buffered saline and dissolved in 0.25 ml of 98% formic acid and incubated for 15 min. The heme content was evaluated by analyzing the clear supernatant at 400 nm, with an extinction coefficient of $1.56 \times 10^5 \times M^{-1} \times cm^{-1}$. The data were normalized to protein content as determined by the Bio-Rad Protein Assay (Bio-Rad).

## Glutathione measurement

Human neurons from patients and controls were incubated with 20 μM ThiolTracker Violet (Invitrogen) for 30 min at 37°C, washed with PBS, and fixed in 4% paraformaldehyde in PBS for 20 min at room temperature. The cells were then permeabilized for 3 min in PBS containing 0.1% Triton X-100, 10% normal goat serum. Next, the cells were incubated with mouse anti-human βIII tubulin (anti-Tuj1, Covance, diluted 1:500) for 1 h at 37°C, and with anti-mouse IgG Alexa Fluor 546 (Immunological Sciences, diluted 1:800). After washing, the cells were randomly analyzed by IN Cell Analyzer 1000 system (GE Healthcare). The ThiolTracker Violet fluorescence in Tuj1-positive cells was collected to compare relative glutathione contents. The quantification of the signal was performed using the NIH image software, ImageJ. A minimum of 100 neurons for each patient and control was analyzed in at least three independent experiments for each sample.

## Determination of ROS

Human neurons were incubated with Alexa Fluor 647 mouse anti-human CD56 (anti-NCAM, BD Biosciences, diluted 1:40) for 1 h, with 20 μM of 2′,7′-dichlorodihydrofluorescein diacetate (H$_2$DCFDA; Molecular Probes) for 15 min, and with 2 μg/ml of Hoechst 33342 for 2 min. All of these incubations were performed at 37°C. The cells were washed and randomly analyzed using an IN Cell Analyzer 1000 system (GE Healthcare). The fluorescence of DCF from NCAM–positive cells was collected to compare the relative ROS contents. The quantification of the signal was performed using the NIH image software, ImageJ. A minimum of 100 neurons for each

patient and control was analyzed in at least three independent experiments for each sample.

## Electron microscopy

Human neurons were fixed in 4% paraformaldehyde and 2.5% glutaraldehyde, post-fixed with 2% $OsO_4$, washed, dehydrated, and embedded in Epon812. Thin sections were stained with uranyl acetate and lead citrate and examined in a Leo912 electron microscope (Zeiss). Images were randomly obtained in blind conditions to the examiner.

## Measurement of dendritic arborization

Labeled neurons were randomly chosen for quantification, with a total of 38 dendritic arborizations analyzed in at least three independent experiments for each sample. Morphometric measurements were made using NeuronStudio image analysis software (http://research.mssm.edu/cnic/tools-ns.html). Individual dendrites were selected randomly and traced manually. The maximum length and branching points were measured and archived automatically.

## Mouse cortical cell cultures

All mice were maintained in accordance with the guidelines established by the European Communities Council (Directive 2010/63/EU of September 22, 2010) and were approved by the local IACUC of the San Raffaele Scientific Institute (Milan, Italy). C57BL/6J female mice were caged in groups of 3–4 animals, while male mice were single-housed and kept on a 12-h light/12-h dark cycle with *ad libitum* access to food and water. Mouse cortical neurons were prepared from E18 mouse embryos as previously described (Dell'Anno *et al*, 2014). E18 mouse cortical neurons were co-cultured with human-derived neurons only for the electrophysiology experiments.

## Patch-clamp electrophysiology

Co-culture experiments of $6 \times 10^4$ cells (half GFP controls and half tdTomato patients) were seeded on Matrigel-coated covers. After 5 days, $2 \times 10^4$ cortical mice neurons were added to improve differentiation and electrophysiological activity. Individual slides containing co-cultured PKAN and control neurons were transferred in a recording chamber mounted on the stage of an upright BX51WI microscope (Olympus, Japan) equipped with differential interference contrast optics (DIC) and an optical filter set for the detection of GFP and tdTomato fluorescence (Semrock, Rochester, NY, USA). Cells were perfused with artificial cerebrospinal fluid (ACSF) containing (in mM): 125 NaCl, 3.5 KCl, 1.25 $NaH_2PO_4$, 2 $CaCl_2$, 25 $NaHCO_3$, 1 $MgCl_2$, and 11 D-glucose, saturated with 95% $O_2$ and 5% $CO_2$ (pH 7.3). The ACSF was continuously flowing at a rate of 2–3 ml/min at room temperature. Whole-cell patch-clamp recordings were performed using glass pipettes filled with a solution containing the following (in mM): 10 NaCl, 124 $KH_2PO_4$, 10 HEPES, 0.5 EGTA, 2 $MgCl_2$, 2 $Na_2$-ATP, 0.02 Na-GTP, (pH 7.2, adjusted with KOH; tip resistance: 4–6 M$\Omega$). All recordings were performed using a MultiClamp 700B amplifier interfaced with a PC through a Digidata 1440A (Molecular Devices). Data were acquired using

**The paper explained**

**Problem**
PKAN, a rare autosomal recessive movement disorder caused by mutations in *PANK2*, belongs to a heterogeneous group of neurodegenerative diseases named neurodegeneration with brain iron accumulation (NBIA). This is a group of disorders that share severe iron overload in specific brain regions, neurodegeneration, and extrapyramidal dysfunction. PKAN is an orphan disorder since no cure to halt or delay the pathological process is currently available. The animal models so far generated do not recapitulate the human brain iron overload.

**Results**
We developed a new PKAN model by generating hiPSC-derived neurons. Patients' neurons present aberrant mitochondria, defective respiration and impaired mitochondrial iron-dependent biosynthesis, heightened ROS levels, failure of action potential firing, and compromised survival. PANK2 reintroduction was sufficient to completely restore some of these alterations. More important, CoA supplementation prevented neuronal death and ROS formation, restoring mitochondrial and neuronal functionality.

**Impact**
Our results indicate that PANK2 deficiency is responsible for abnormal phenotypes in human neuronal cells and highlight CoA treatment as a potential therapeutic option.

pClamp10 software (Molecular Devices) and analyzed with GraphPad Prism 5 and SigmaStat 3.5 (Systat Software Inc.). Voltage- and current-clamp traces were sampled at a frequency of 10 kHz and low-pass filtered at 2 kHz. The input resistance ($R_{in}$) was calculated by dividing the steady-state voltage response to a negative current step ($-10$ to $-50$ pA, 1 s) by the amplitude of the injected current. Labeled GFP or tdTomato neurons were randomly chosen for measurement, and no blind experiments were done for electrophysiology studies.

## Determination of respiratory activity

Oxygen consumption rate (OCR) was measured in PKAN and control neurons with a XF96 Extracellular Flux Analyzer (Seahorse Bioscience, Billerica, MA, USA). Each control and PKAN NPC was seeded on a XF 96-well cell culture microplate (Seahorse Bioscience) at a density of $15–20 \times 10^3$ cells/well and differentiated as previously described. After replacing the growth medium with 180 μl of bicarbonate-free DMEM pre-warmed at 37°C, cells were incubated at 37°C without $CO_2$ for 1 h before starting the assay procedure. Then, baseline measurements of OCR, after addition of 1 μM oligomycin and of 2.1 μM carbonyl cyanide 4-(trifluoro-methoxy) phenylhydrazone (FCCP), were measured using an already established protocol (Invernizzi *et al*, 2012). Data were expressed as pmol of $O_2$ per minute and normalized by cell number measured by the CyQUANT Cell proliferation kit (Invitrogen), which is based on a fluorochrome binding to nucleic acids. Fluorescence was measured in a microplate luminometer with excitation wavelength at $485 \pm 10$ nm and emission detection wavelength at $530 \pm 12.5$ nm. All measurements were performed in 24 replicates for each sample. At least three different experiments were carried out in different days. Experiments were carried out in blind conditions to the examiner.

## Statistical analyses

Statistical methods were not employed to predetermine sample size in the experiments. All the experiments were performed at least in triplicate; data were analyzed using GraphPad Prism. In general, for normally distributed data, two-tailed unpaired Student's *t*-test, *z*-test, and one- or two-way ANOVA followed by Bonferroni post-test were used. For non-normally distributed data, Mann–Whitney rank sum test was used. Shapiro–Wilk test was used to test for normal distribution. The data are reported as the mean + SEM. The *P*-value < 0.05 was considered statistically significant, and the exact value is reported in the figures.

Expanded View for this article is available online.

## Acknowledgements
The financial support from Telethon-Italia (Grants no. GGP11088 to SL and VT, GGP16234), AISNAF (to SL), Italian Ministry of Health (RF10-20 to VB and GR11-04 to SGG), European Research Council (AdERC #340527 to V.B.), TIRCON project (FP7/2007-2013, HEALTH-F2-2011, # 277984 to VT), and Mariani Foundation of Milan is gratefully acknowledged. Part of this work was carried out in ALEMBIC, an advanced microscopy laboratory established by the San Raffaele Scientific Institute and the Vita-Salute San Raffaele University. We thank the Cell Line and DNA Bank of Pediatric Movement Disorders and Mitochondrial Diseases of the Telethon Genetic Biobank Network (project no. GTB07001) and the Bank for the Diagnosis and Research of Movement Disorders (MDB) of the EuroBiobank. We thank Dr. Daniel Morgan for editing the manuscript.

## Author contributions
DO and AR developed the neuronal models; PS and DO performed analysis on neurons; LY performed electrophysiological recordings; SGG generated and produced PANK2-expressing lentiviruses; CC and PGM established and maintained hiPSCs; MF provided the genetic analysis of hiPSC; AC performed biochemical experiments on neurons, SD and PV performed genetic and microscale oxygraphy analysis; BG provided fibroblasts from the biobank; ST analyzed electrophysiological data and wrote the manuscript; SL, VB, VT conceived the study and wrote the manuscript.

## Conflict of interest
The authors declare that they have no conflict of interest.

## For more information
OMIM: http://www.ncbi.nlm.nih.gov/omim/
U.S.A. Patients Association: www.nbiadisorders.org/
Orphanet: www.orpha.net/consor/cgi-bin/OC_Exp.php?Lng=IT&Expert=385
Italian Patients Association: AISNAF: aisnaf.org/

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
