## [Review Process File · EMBO Molecular Medicine]

Coenzyme A corrects pathological defects in human neurons of PANK2-associated neurodegeneration

Daniel Orellana, Paolo Santambrogio, Alicia Rubio, Latefa Yekhlief, Cinzia Cancellieri, Sabrina Dusi, Serena Giannelli, Paola Venco, Pietro Mazzara, Anna Cozzi, Maurizio Ferrari, Barbara Garavaglia, Stefano Taverna, Valeria Tiranti, Vania Broccoli, and Sonia Levi

Corresponding author: Sonia Levi, Vita-Salute San Raffaele University

Review timeline:

Submission date:	09 March 2016
Editorial Decision:	31 March 2016
Revision received:	01 June 2016
Editorial Decision:	30 June 2016
Revision received:	13 July 2016
Accepted:	25 July 2016

Transaction Report:

Editor: Celine Carret

1st Editorial Decision

31 March 2016

Thank you for the submission of your manuscript to EMBO Molecular Medicine. We have now heard back from the two referees whom we asked to evaluate your manuscript. Although the referees find the study to be of potential interest, they also raise a number of concerns that need to be addressed in the next final version of your article.

As you will see from the comments below, the two referees are rather positive about the study but do have suggestions and recommendations to further improve conclusiveness and clarity as well as increase the potential clinical impact, which is particularly important for our scope.

I will not get into experimental details, but we feel that the referees' reports are very clear and nicely detailed and we would like to encourage you to address all issues raised as recommended.

Given these evaluations, I would like to give you the opportunity to revise your manuscript, with the understanding that the referees' concerns must be fully addressed and that acceptance of the manuscript would entail a second round of review. Please note that it is EMBO Molecular Medicine policy to allow only a single round of revision and that, as acceptance or rejection of the manuscript will depend on another round of review, your responses should be as complete as possible.

I look forward to receiving your revised manuscript.

***** Reviewer's comments *****

Referee #1 (Comments on Novelty/Model System):

All my suggestions are included in the "remarks sent to the author"

Referee #1 (Remarks):

The present study focuses on the rare disorder "pantothenate kinase-associated neurodegeneration" ("PKAN"). The authors have reprogrammed PKAN patient and control fibroblasts and ultimately generated iPSCs-derived forebrain neurons. iPSC-derived cells were studied at the stage of neuronal precursors but also at the stages of short vs. long-term differentiated neurons. The generated cells were characterized in terms of cellular identity, morphology and electrophysiological properties. In addition PKAN-related alterations were described in terms of load of reactive oxygen species, mitochondrial morphology and iron processing machineries. The authors have previously used similar approaches and published cellular phenotypes (emphasizing in oxidative status and iron-handling properties) exhibited by PKAN patient-derived skin fibroblasts but also, most recently, neurons generated by direct reprogramming.

Key elements

1. Generation and characterization of iPSC-derived PKAN-related neurons.
2. Patient-related neurons do not have an altered cellular morphology, but do exhibit altered electrophysiological properties.
3. Long term neuronal cultures from patients show less number of cells than those of controls.
4. PKAN-related neurons have abnormal mitochondrial morphology, reduced mitochondrial membrane potential, increased ROS production, reduced glutathione levels, and decreased oxygen consumption.
5. Iron-related machinery is impaired in patient-related neuronal precursor cells.
6. Addition of CoA partially restores cell viability, alleviates ROS load, and increases oxygen consumption rate in patient-related neurons.

The authors aim to decipher the molecular mechanisms by which iron metabolism defects in PKAN disease may be linked to mitochondrial dysfunction in a human iPSC-derived neuronal model of the disease. The rationale and experimental strategy are relevant and the objective is of great scientific importance. However, this is an exclusively in vitro study and the cells used do not have a well characterized and, more importantly, disease-relevant neuronal phenotype. Although the overall quality of the manuscript is good, the weakness of this in vitro study is that it is mostly descriptive and provides no novel disease-related mechanistical insight.

Although the results are promising, several points remain to be conclusively addressed. In addition, the discussion should be reconstructed.

Major concerns

1- Insufficient characterization of the iPSC-derived neuronal subtype of the study

- a) Data covering age, gender, and comorbidities of control and patient donors must be provided.
- b) The ICC data concerning markers of pluripotency and differentiation should be demonstrated side-by-side for control and patient-derived cells.
- c) The authors refer to their model as having a forebrain-related glutamatergic phenotype. Data is not sufficient as there is no quantification for the % of cells expressing any marker relevant to such an acclaimed phenotype. For instance, the representation of MAP2+ and vGLUT+ cells as opposed

to GABAergic subtypes should be shown.

d) DCX must be quantified for control and patient lines at the stage of neural rosettes, in order to assess the neuronal differentiation potential of the different lines. Otherwise authors cannot state in the discussion (p.12 -13):

The first result of our work is the successful generation of PKAN hiPSCs and their neuronal derivatives, suggesting that PANK2 deficiency does not affect the neuronal fate commitment [...].

e) Tuj1 is a mature neuronal marker (see for reference Marcetto et al., 2010) and should not be expressed by NPCs and especially not be used as a NPC marker (Figure EV3C). Especially since authors themselves use it as a maturation marker later on:

[...] and PKAN neurons appeared to have developed complex morphology, organized a dense network and expressing crucial neuronal markers like Tuj1, MAP2, and NeuN (Fig. 1B). (p. 7)

2- Use of mouse cortical neurons (?)

In the Materials and Methods, the authors claim that they used mouse neurons to enhance neuronal differentiation (p. 17). The exact figures which represent such co-cultures must be specified. Where were mouse neurons derived/ obtained from? Did authors use human specific antibodies for immunocytochemistry and western blots to differentiate between mouse and human derived proteins?

3- WB assessment of PANK2 abundance

The blots concerning PANK2 in Figures 1 and 3C are of poor quality. Firstly, in Figure 1C the pattern of this protein in Control 3 is significantly different than in the other 2 Control lines. Also, the unspecific "*" band appears to be up-regulated in patient lines. The authors should optimize the blot to better separate the specific from the unspecific form of the protein and it would be good to use an antibody isotype (if possible) to convincingly show that the * band is unspecific. Secondly, in Figure 3, the authors should provide WB data showing PANK2 in control and patient lines, with and without PANK2 transduction, on the same blot.

4- Insufficient characterization of mitochondrial homeostasis.

Striking ultrastructural differences are reported in Figure 2B, yet they are not accompanied by a sufficient analysis of markers relevant to mitochondrial mass, morphology or the abundance of respiratory chain complex components. Such an approach is important and would significantly add to the justification and interpretation of abnormal mitochondrial morphology. Moreover, the decreased oxygen consumption and the changes in the intensity of the dye used to determine mitochondrial membrane potential can be strongly influenced by differences in mitochondrial mass and volume.

5- "Loading" of WB in Figure 4

The word "loading" must be replaced with the exact name of the protein or means by which loading was controlled and used for quantification of the results.

6- Inconsistency of patient lines used

According to the current version, the use of patient lines is not consistent throughout manuscript. Different experiments were conducted in different lines. In Figure 4B the lines PKAN (F419fsX472a)#5 and PKAN (F419fsX472b)#11 are used, whereas in Figure 4C the lines #3 are used for both cases. If these numbers refer to different iPSC clones generated from the same individual this must be described in more detail. The nomenclature is also inconsistent and unless this issue is clarified, the conclusions of the study are not justified by the data, as presented. Even more so, solely 1 patient group is reported for Figure 5, which is addressing an interventional approach.

7- Mitochondrial dysfunction and aberrant iron-related machinery are poorly linked

A major point of the study is the potential link between mitochondrial dysfunction, oxidative stress and iron metabolism. The authors should experimentally address whether indeed the increased ROS

load is due to mitochondria dysfunction. The study of Krebs cycle derivatives would also be of relevance here. Following, there is no conclusive experimental proof showing the link between aberrant ferritin and TfR levels and mitochondrial dysfunction and this point, although made, is not sufficiently discussed. An in-depth look into both pathways and the mode they may be linked with each other is important, especially for the subsequent rationale of using CoA as an interventional approach.

8- Effect of CoA administration to cells

The use of CoA as a therapeutic intervention *in vivo* has obvious limitations, the first being that it will lead to systemic effects. In the present study, exogenous addition of CoA is used and partial restoration of PANK2-related mitochondrial phenotypes is reported in one single case. This is not sufficient. Furthermore, it is important to test this effect in a concentration-dependent manner. Also, the authors must address the effects on the oxygen consumption rate of control cells as well. Last, the authors may consider to address (again similar to points 4 and 5) how mitochondria are influenced at an ultrastructural level.

Additional concerns and suggestions

- Globus pallidus and substantia nigra are two brain regions which are primarily affected by the PKAN disease and show iron accumulation. It would be interesting to complement this study by driving the generation of neurons more biochemically relevant to such brain regions.
- Lack of iron deposition in this system confounds the investigation of molecular processes underlying PKAN disease. The data presented, nicely suggests increased levels of TfR and reduced ferritin. But is this ultimately related to iron deposition *in vivo*? The authors could for instance address whether in their system TfR trafficking is intact, in order to first explain but also enforce the relevance of their model.
- The authors may consider the use of CCCP and Oligomycin in order to define the specificity the mitochondrial potential staining used in the present study.
- Although in Figure 2 the size of mitochondria is depicted in a graph, the pictures of patient line mitochondria are strikingly larger than the control ones. The phenotype is convincing, given the graph accompanying the observation, nevertheless the choice of more representative pictures or a larger frame is more suitable.
- In Figure 2D, there should be a clearer graphical depiction of the statistical comparisons made.
- The use of "control" should be carefully revised as it is confusing the reader. Control cells or Control condition?
- T-test is inappropriate for Figure 3A, since more than 3 conditions per group were studied.
- When describing the results of the rescue experiment in Figure 3A, authors refer to the wrong Figure number: As expected, PANK2 re-expression, confirmed by immunoblotting (Fig. 3C), was sufficient to reduce ROS levels in PKAN neurons (Fig. 3C). (p. 10) The quality of graph E in Figure 5 should be improved.
- Literature format is in a few cases not uniform.
- Grammar should be carefully revised.
- Reconsider the use of "-sensitive" instead of "-sensible"

Recommendation:

The generation of an iPSC-derived and long-term cultivated neuronal model of this rare disease is very important. The generation of the model is *per se* a milestone, yet the study provides little novelty concerning the *in vivo* disease-related phenotype and also the mechanism underlying the amelioration of neurodegeneration after coenzyme A administration. The tackling of the mechanism may further be limited by the fact that the study was conducted in a neuronal subtype which is not sufficiently characterized in the study and does not exhibit the disease-related iron deposition.

Referee #2 (Comments on Novelty/Model System):

The novelty and medical impact are high as the authors generated iPSC-derived neurons of PKAN2 patients that represent an appropriate model for neurological disease. However, some points have to be precised (see Remarks to be sent to the authors)

Referee #2 (Remarks):

The manuscript by Orellana and co-workers reports the characterization of human iPSC-derived neurons of PKAN2 patients. The authors generated iPSC and their neuronal derivatives using cell lines from controls and patients with PKAN2 mutations. Next, they showed that PKAN neurons present abnormal oxygen consuming rate, increased ROS overproduction and abnormal electrophysiological properties; PKAN neurons also displayed impaired iron metabolism. Finally, the authors show that CoA supplementation corrects the pathological defects in PKAN neurons. Overall, this cellular model recapitulates earlier observations in patient's fibroblasts that were previously reported by this group. However, the generation of iPSC-derived neurons of PKAN2 patients allowed or will allow to investigate specific functions not present in cultured skin fibroblasts.

The manuscript is very interesting, the iPSC and neurons are well characterized. Nevertheless, I have several questions.

The three PKAN patients have either a PANK2 premature stop codon or frameshift mutations that result in the complete lack of PANK2 protein. This is clearly shown by the Western blot presented in Figure 1C that shows the absence of PANK2 protein in the three patients compared to the three controls. Why are the PANK2 signal and the non-specific band in controls 1 and 2 so light compared to control 3 and the three patients while the loading controls seems to be relatively similar in all samples? The authors clearly show that PKAN neurons have abnormal firing activity in Figure 1F. However, there are no standard deviations in Figure 1G. These should be added.

The authors detected increased ROS production in PKAN neurons that is reduced by re-expression of wild-type PKAN2. Re-expression of PANK2 was confirmed by Western blot that shows the presence of the normal PANK2 protein in the iPSC-derived neurons of the three patients. In Figure 3C the signal for control 1 cells is much stronger than in Figure 1C. The previously observed non-specific band is present only in control cells, but not in patients' cells which present this non-specific band in Figure 1C. Moreover, the three patients cell lines also present an additional band closed to the 62 kDa PANK2 band. Does this band correspond to the mature form of the mitochondrial protein after cleavage of the MTS? The authors should comment on this band and the disappearance of the non-specific band.

Decreased activities of cytosolic and mitochondrial aconitases were observed in PKAN neurons by in-gel activities presented in Figure 4a. The loading in this figure is not optimal especially for the PKAN (Y190X)#1 and the PKAN (F419fsX427a)#5 cells. For this reason and the presence of a high background, the c.ACO signal is difficult to evaluate. Moreover, two different blots are presented: one with three controls and the first two patients and another one with control 1 and the third patient. This doesn't allow their comparison in the histograms. The authors should present histograms related to a single gel containing all samples. Conversely, there is no quantification of the mitochondrial and cytosolic aconitase proteins (Figure 4B). This should be done, as the loading is somewhat different for the various samples. Moreover, the actin signal is very strong and the blot seems overexposed.

The authors supplemented PKAN neurons with CoA and observed increased cell viability and increased heme content after 150 days of treatment. They also studied the effect of CoA supplementation in PKAN neurons on disease-associated ROS overproduction and on their functional properties (Figure 5C), but the effect of CoA supplementation on control cells was not evaluated. This should be included.

Responses to Reviewer's comments

We wish to thank all the referees for their careful reading, positive comments, and helpful suggestions to our manuscript.

Below is a point-by-point reply to the referees' comments:

Referee #1

Major concerns.

1- Insufficient characterization of the iPSC-derived neuronal subtype of the study.

a) Data covering age, gender, and comorbidities of control and patient donors must be provided.

We agree with the referee's suggestion and we have now added to the Materials and Methods section ("Fibroblasts culture and hiPSC generation" chapter, page 17) the age, gender, and comorbidities of patient donors. Furthermore, In addition, we have now explained that two controls used in our experiments came from two neonatal male control fibroblasts purchased from ATCC and one female come from cord blood stem cells.

b) The ICC data concerning markers of pluripotency and differentiation should be demonstrated side-by-side for control and patient-derived cells.

We agree with the referee's suggestion and we apologize for the previous figure. We have now prepared a new **Figure EV3 B and C** where we compare side-by-side control and patient rosettes and NPCs.

c) The authors refer to their model as having a forebrain-related glutamatergic phenotype. Data is not sufficient as there is no quantification for the % of cells expressing any marker relevant to such an acclaimed phenotype. For instance, the representation of MAP2+ and vGLUT+ cells as opposed to GABAergic subtypes should be shown.

We are grateful to the reviewer for the interesting point raised. We have now prepared a new **Figure EV3 B** where we have added new IF and quantifications for the forebrain markers Tbr2 and Ctip2 (Ricciardi et al 2012) expressed in neuronal rosettes of control and PKAN patients with equal intensity.

FoxG1 and Pax6 (also forebrain markers) were not quantified since almost all the cells expressed these markers with equal intensity. On page 7 we modified the manuscript accordingly (lines 2-5 in red).

Additionally, new experiments were performed showing the percentage of double VGLUT1/Map2, TH/Map2 and GABA/Map2 positive neurons, **Figure EV 4B**.

d) DCX must be quantified for control and patient lines at the stage of neural rosettes, in order to assess the neuronal differentiation potential of the different lines. Otherwise authors cannot state in the discussion (p.12 -13):

The first result of our work is the successful generation of PKAN hiPSCs and their neuronal derivatives, suggesting that PANK2 deficiency does not affect the neuronal fate commitment [...].

We have added a quantification of the DCX positive cells in **Figure EV 3B**.

e) Tuj1 is a mature neuronal marker (see for reference Marcetto et al., 2010) and should not be expressed by NPCs and especially not be used as a NPC marker (Figure EV3C). Especially since authors themselves use it as a maturation marker later on:

[...] and PKAN neurons appeared to have developed complex morphology, organized a dense network and expressing crucial neuronal markers like Tuj1, MAP2, and NeuN (Fig. 1B). (p. 7)

We apologize for this inconvenient. As correctly stated by the reviewer TuJ1 is an early and specific marker of post-mitotic neurons and should not be used to indicate NPCs. A small fraction of NPCs can spontaneously differentiate into neurons even in proliferative conditions and this explains the pictures we initially reported. In the new **Figure EV 3C** this picture has been eliminated and

replaced with NPCs immunodecorated with Nestin a specific marker for undifferentiated NPCs and the text changed accordingly

However, the Tuj1 marker for the early neuron remains in **Figure 1B**.

2- Use of mouse cortical neurons (?)

In the Materials and Methods, the authors claim that they used mouse neurons to enhance neuronal differentiation (p. 17). The exact figures which represent such co-cultures must be specified. Where were mouse neurons derived/ obtained from? Did authors use human specific antibodies for immunocytochemistry and western blots to differentiate between mouse and human derived proteins?

We have specified in the revised manuscript that the co-culture of human neurons with embryonic day-18 cortical mice neurons were only used for the electrophysiology experiment (pag 8, lane 13). They can be found also in Materials and Methods section, "Patch-clamp electrophysiology" chapter, page 23. Besides we have now included in the Materials and Methods section (on page 22) the method used to obtain mouse cortical neurons.

3- WB assessment of PANK2 abundance

The blots concerning PANK2 in Figures 1 and 3C are of poor quality. Firstly, in Figure 1C the pattern of this protein in Control 3 is significantly different than in the other 2 Control lines. Also, the unspecific "" band appears to be up-regulated in patient lines. The authors should optimize the blot to better separate the specific from the unspecific form of the protein and it would be good to use an antibody isotype (if possible) to convincingly show that the * band is unspecific. Secondly, in Figure 3, the authors should provide WB data showing PANK2 in control and patient lines, with and without PANK2 transduction, on the same blot.*

We have optimized the blot for PANK2 in neurons and reported the results in **Figure 1C**. This figure now contains: controls, PKAN, and PKAN-transduced neurons in the same blot. The unspecific band is present in all tested samples. We do not know the nature of this band, but we assigned the title of unspecific band because it is reported also in the antibody data sheet (anti-human PANK2 clone 3H9 antibody, Origene). In addition, as the same antibody was used in the IF in **Figure 1A** and no signal was detected in the PKAN-derived NPCs. Moreover, our group has recently demonstrated that, in pure mitochondria sub-cellular fractions from mice brain, only one band was detected by the same antibody (Brunetti et al 2012).

For these reasons we do not believe that the lower band corresponds to the mature form of mitochondrial protein after cleavage.

4- Insufficient characterization of mitochondrial homeostasis.

Striking ultrastructural differences are reported in Figure 2B, yet they are not accompanied by a sufficient analysis of markers relevant to mitochondrial mass, morphology or the abundance of respiratory chain complex components. Such an approach is important and would significantly add to the justification and interpretation of abnormal mitochondrial morphology. Moreover, the decreased oxygen consumption and the changes in the intensity of the dye used to determine mitochondrial membrane potential can be strongly influenced by differences in mitochondrial mass and volume.

We are grateful to the reviewer for the interesting point raised. Unfortunately, it was not possible for us to investigate markers of mitochondrial mass within the time frame allowed for the revision. This is mainly due to the low yield of mitochondria extracts obtained from our neurons cultures, which allowed us to perform only one experiment without any replicates. So, we are not confident about reliability of results, which need to be confirmed. We agree that changes in mitochondrial mass could influence the mitochondrial membrane potential (MMP) and oxygen consumption. However, our analyses clearly show an increase in mitochondrial size in PKAN derived neurons, implicating possibly an increase in mitochondrial mass, while the MMP and oxygen consumption are significantly decreased.

5- "Loading" of WB in Figure 4

The word "loading" must be replaced with the exact name of the protein or means by which loading was controlled and used for quantification of the results.

We did not explain this concept in a correct way. Figure 4A corresponds to an in-gel enzymatic activity, it is not a western blot. We now added a clear explanation in the figure legend (**Figure 4A**) indicating that the lower part of the gel was cut and stained with Coomassie blue, to verify equally loaded proteins amount. Additionally, we now named **Figure 4A** as "In-gel enzymatic activity" to avoid any misleading and clearly described the figure's content.

6- Inconsistency of patient lines used

According to the current version, the use of patient lines is not consistent throughout manuscript. Different experiments were conducted in different lines. In Figure 4B the lines PKAN (F419fsX472a)#5 and PKAN (F419fsX472b)#11 are used, whereas in Figure 4C the lines #3 are used for both cases. If these numbers refer to different iPSC clones generated from the same individual this must be described in more detail. The nomenclature is also inconsistent and unless this issue is clarified, the conclusions of the study are not justified by the data, as presented. Even more so, solely 1 patient group is reported for Figure 5, which is addressing an interventional approach.

We now included a more detailed description in the revised manuscript. We stated that we used three clones for each one of the three patients (identified with different #) in page 6. The experiments were done for all the different clones and one clone for each patient is indicated as a representative example in the figures. The nomenclature was modified to make it clear and consistent throughout the manuscript.

In Figure 5 "PKAN patients" is intended to include the three PKAN derived neurons from the three PKAN patients represented together and not solely a patient group. In the old version only **Figure 5D** included one patient. This has been improved, as we were able to repeat the experiments with the three patient's derived neurons. **Figure 5D** now includes an analysis of the oxygen consumption rate and the recovery after CoA treatment in the three PKAN patients.

7- Mitochondrial dysfunction and aberrant iron-related machinery are poorly linked

A major point of the study is the potential link between mitochondrial dysfunction, oxidative stress and iron metabolism. The authors should experimentally address whether indeed the increased ROS load is due to mitochondria dysfunction. The study of Krebs cycle derivatives would also be of relevance here. Following, there is no conclusive experimental proof showing the link between aberrant ferritin and TfR levels and mitochondrial dysfunction and this point, although made, is not sufficiently discussed. An in-depth look into both pathways and the mode they may be linked with each other is important, especially for the subsequent rationale of using CoA as an interventional approach.

We chose to evaluate ROS development using cytosolic DCF instead of mitochondrial specific Rhodamine-123 for technical reasons. For the latter reagent, the detection of fluorescent Region of Interest (ROI) was not reliable in our experimental setting (IN Cell Analyzer). In addition, we obtained similar results with the two reagents in our previous experiments described in Santambrogio et al, 2015.

Concerning the study of Krebs cycle derivatives, we are grateful to the reviewer for the interesting point raised. Even if these suggestions are very interesting and helpful for future work, unfortunately we have no available data to fulfill this request at the moment. Indeed, we are currently setting up the condition to measure the Krebs cycle derivatives by mass spectroscopic methods, however our present data are not conclusive and we need to perform other experiments that will require several months of work. In any case, we have implemented the discussion section with our hypothesis on the possible link between ISC deficiency, energy impairment and iron dysregulation (pag. end 14-15).

8- Effect of CoA administration to cells

The use of CoA as a therapeutic intervention in vivo has obvious limitations, the first being that it will lead to systemic effects. In the present study, exogenous addition of CoA is used and partial restoration of PANK2-related mitochondrial phenotypes is reported in one single case. This is not sufficient. Furthermore, it is important to test this effect in a concentration-dependent manner. Also, the authors must address the effects on the oxygen consumption rate of control cells as well. Last, the authors may consider to address (again similar to points 4 and 5) how mitochondria are influenced at an ultrastructural level.

We agree with the referee's suggestion. We performed additional experiments of external CoA administration in all the three PKAN patients and we evaluated several parameters including: firing activity, ROS production, Oxygen consumption and heme content. Moreover, we also verified the effect of CoA administration also in controls' neurons. All these data are now present in the new version of the manuscript. However, we could not manage to repeat all the experiments testing different CoA concentration within the time allotted for the revision process. We based our decision of using 25 microM of CoA on previously published data (Srinivasan et al, 2015). In fact, Srinivasan and coworkers tested different concentrations of CoA and established that 25 microM of exogenous CoA administration was effective in reverting abnormal phenotypes in HEK293 and Drosophila S2 chemicals cells models of PKAN.

We also managed to carry out experiments using PKAN patients neurons transduced with PANK2. In the previous version, only ROS levels were reported to be restored with PANK2 addition. We now present additional data on mitochondria abnormal morphology and oxygen consuming rate after PANK2 re-expression. PANK2 reverted the increase in mitochondrial size and aberrant morphology in all the three PKAN patient neurons. This is now presented in a new graph in **Figure 2B**. PANK2 re-expression also reverted the decrease in oxygen consuming rate for the three PKAN patient neurons, which was statistically significant only in the basal and FCCP conditions but not in the oligomycin condition. This is now presented in a new graph in **Figure 2D**.

Additional concerns and suggestions

- Globus pallidus and substantia nigra are two brain regions, which are primarily affected by the PKAN disease and show iron accumulation. It would be interesting to complement this study by driving the generation of neurons more biochemically relevant to such brain regions.

We agree with the referee's suggestion. Recently, a protocol to generate efficiently high quality GABAergic neurons derived from human iPSCs was published (Colasante et al, 2015). We are planning to use this protocol with our PKAN iPSCs in order to obtain a more suitable model, which implicates a complete new project. Unfortunately, we currently lack the financial support to carry out both of them in parallel.

- Lack of iron deposition in this system confounds the investigation of molecular processes underlying PKAN disease. The data presented, nicely suggests increased levels of TfR and reduced ferritin. But is this ultimately related to iron deposition in vivo? The authors could for instance address whether in their system TfR trafficking is intact, in order to first explain but also enforce the relevance of their model.

We agree with the referee's suggestion. We expected iron accumulation in our PKAN neuronal model. However no evidence of iron accumulation was observed in our analysis. These could be due to different reasons: 1) it may be possible that a longer differentiation time will be necessary to observe clear iron overload; 2) the glutamatergic neuronal cell type obtained with our method does not represent the major neuronal types found in the globus pallidus and substantia nigra (GABAergic and dopaminergic neurons, respectively); 3) the main cells types accumulating iron with age are microglia and astrocytes. It is possible that in co-culture conditions of PKAN derived neurons with PKAN derived microglia or astrocytes a clear iron deposition in the neurons would be more evident. We are planning to answer to all these points in future work.

We agree with the referee's that the study of TfR1 trafficking could be significant. Indeed, we standardized a methodology to test the TfR1 trafficking in PKAN fibroblast with the idea to test it later on PKAN derived neurons. The method consists in testing the neurons a 4°C after addition of

labeled Transferrin. But it is technically difficult to do because the signal on neurons is not so easy to interpret and we consider it not be performed in the available time for the revision.

- The authors may consider the use of CCCP and Oligomycin in order to define the specificity the mitochondrial potential staining used in the present study.

We now present IF data demonstrating that the TMRM signal comes specifically from mitochondria. We treated control and PKAN neurons with the mitochondria uncoupler FCCP. The data are shown in the new **Figure 2A**.

- Although in Figure 2 the size of mitochondria is depicted in a graph, the pictures of patient line mitochondria are strikingly larger than the control ones. The phenotype is convincing, given the graph accompanying the observation, nevertheless the choice of more representative pictures or a larger frame is more suitable.

We have now added new images of more representative pictures in **Figure 2B**.

- In Figure 2D, there should be a clearer graphical depiction of the statistical comparisons made.

We now added a clear graphical depiction of the statistical comparisons of the figure, that now is showed in **Figure 2C**.

- The use of "control" should be carefully revised as it is confusing the reader. Control cells or Control condition?

We have corrected the previous Figure 2D, now Figure 2C.

- T-test is inappropriate for Figure 3A, since more than 3 conditions per group were studied.

We have performed a one-way ANOVA test and modified **Figure 3A** and legend accordingly.

- When describing the results of the rescue experiment in Figure 3A, authors refer to the wrong Figure number: As expected, PANK2 re-expression, confirmed by immunoblotting (Fig. 3C), was sufficient to reduce ROS levels in PKAN neurons (Fig. 3C), (p. 10) The quality of graph E in Figure 5 should be improved.

We have modified **Figure 3C**. The previous blot in **Figure 3C** was eliminated and a new blot for PKAN that now include control, PKAN patients, and PKAN transduced neurons is included in **Figure 1C** as required by Referee 2.

- Literature format is in a few cases not uniform.

We have thoughtfully revised the references' format to uniform it.

- Grammar should be carefully revised.

We have thoroughly revised the grammar and corrected mistyping.

- Reconsider the use of "-sensitive" instead of "-sensible"

Thank for this observation. We have included “-sensitive” in the legends of Figure 2, 3, and 5.

Referee #2

The three PKAN patients have either a PANK2 premature stop codon or frameshift mutations that result in the complete lack of PANK2 protein. This is clearly shown by the Western blot presented in Figure 1C that shows the absence of PANK2 protein in the three patients compared to the three controls. Why are the PANK2 signal and the non-specific band in controls 1 and 2 so light

compared to control 3 and the three patients while the loading controls seems to be relatively similar in all samples?

We agree with the referee's suggestion. We have optimized the blotting for PANK2 in Figure 1C and we have eliminated the blot presented in Figure 3C and now added a new **Figure 1C**. This figure contains: controls, PKAN, and PKAN-transduced neurons in the same blot. The unspecific band is present in all tested samples. We do not know the nature of this band, but we assigned the title of unspecific band because it is reported also in the antibody data sheet. In addition, as previously stated the same antibody was used in the IF in **Figure 1A** and no signal was detected in the PKAN-derived NPCs. Moreover, our group has recently demonstrated that, in pure mitochondria sub-cellular fractions from mice brain, only one band was detected by the same antibody (Brunetti et al 2012).

For these reasons we do not believe that the lower band corresponds to the mature form of mitochondrial protein after cleavage.

The authors clearly show that PKAN neurons have abnormal firing activity in Figure 1F. However, there are no standard deviations in Figure 1G. These should be added.

We have modified the figure adding the respective SEM.

The authors detected increased ROS production in PKAN neurons that is reduced by re-expression of wild-type PKAN2. Re-expression of PANK2 was confirmed by Western blot that shows the presence of the normal PANK2 protein in the iPSC-derived neurons of the three patients. In Figure 3C the signal for control 1 cells is much stronger than in Figure 1C. The previously observed non-specific band is present only in control cells, but not in patients' cells which present this non-specific band in Figure 1C. Moreover, the three patients cell lines also present an additional band closed to the 62 kDa PANK2 band. Does this band correspond to the mature form of the mitochondrial protein after cleavage of the MTS? The authors should comment on this band and the disappearance of the non-specific band.

We thank the referee for this suggestion that was also raised by referee#1. We respond to this issue in the first point.

Decreased activities of cytosolic and mitochondrial aconitases were observed in PKAN neurons by in-gel activities presented in Figure 4a. The loading in this figure is not optimal especially for the PKAN (Y190X)#1 and the PKAN (F419fsX427a)#5 cells. For this reason and the presence of a high background, the c.ACO signal is difficult to evaluate. Moreover, two different blots are presented: one with three controls and the first two patients and another one with control 1 and the third patient. This doesn't allow their comparison in the histograms. The authors should present histograms related to a single gel containing all samples. Conversely, there is no quantification of the mitochondrial and cytosolic aconitase proteins (Figure 4B). This should be done as the loading is somewhat different for the various samples. Moreover, the actin signal is very strong and the blot seems overexposed.

Unfortunately, the measurement of the enzymatic activity of aconitases requires a large number of neurons and is detectable only on homogenates from freshly harvested cells, thus it is not easy to obtain sufficient neurons of all the three controls and all the three patients simultaneously. For this technical reason, we are showing two gels that are clearly distinct, as allowed by the editorial policy.

The quantification of the mitochondrial and cytosolic aconitase proteins have been added to the new version of **Figure 4B**.

The authors supplemented PKAN neurons with CoA and observed increased cell viability and increased heme content after 150 days of treatment. They also studied the effect of CoA supplementation in PKAN neurons on disease-associated ROS overproduction and on their functional properties (Figure 5C), but the effect of CoA supplementation on control cells was not evaluated. This should be included.

We thank the referee for this suggestion. We have now added the respective control cells treated with CoA to different experiments reported in Figure 5, specifically panel C showing ROS measurement, panel D showing oxygen consuming rate, and panel E showing heme content.

2nd Editorial Decision

30 June 2016

Thank you for the submission of your revised manuscript to EMBO Molecular Medicine. We have now received the enclosed reports from the referees that were asked to re-assess it. As you will see, while referee 2 is happy, referee 1 remains very much concerned with issues that preclude acceptance right now. However, as we do find that the paper would be a good fit for EMBO Molecular Medicine, we have asked for additional editorial advice and as a result, we would like to give you a chance to revise your article one final time along the lines given below. From referee 1 report, please absolutely address the following:

- 1- There is significant inconsistency regarding the stage of differentiation during which electrophysiological recordings, mitochondrial mass, and cell survival have been assessed.
- 2- The authors must address the stage of differentiation at which neuronal loss is observed. Are significant changes in neuron number at d 120 already present? This is important to evaluate the beneficial effect of CoA.
- 3- In Figure 5B, information on the repetitively firing neurons with and without CoA should be given for both control and patient lines.
- 4- The authors must clarify which patient lines and clones were used for electrophysiological recordings and mitochondria ultrastructural analysis.

Minor concerns:

- 5) Figure legends are extensive
- 6) Language still needs to be thoroughly revised in several parts of the manuscript.

Referee 2 concerns must be equally addressed.

Please submit your revised manuscript within 3 weeks. I look forward to seeing a revised form of your manuscript as soon as possible.

***** Reviewer's comments *****

Referee #1 (Remarks):

In their revised manuscript, Orellana et al have included additional data which sufficiently cover some of the major concerns previously raised. More specifically, a) the characterization of the generated iPSC lines, neural precursor lines, and iPSC-derived neurons and b) information about the control and patient donors has been incorporated. In addition, c) the use of mouse cortical neurons for electrophysiological recordings is explained and d) Figure 5 now includes results from cultures corresponding to three patients. Collectively these additions have improved the quality of the manuscript to some extent. However, these additions are not sufficient providing solid insights into the iron-related phenotype in their model. More importantly, the described phenotypes related to mitochondrial morphology, oxygen consumption, oxidative stress, and iron metabolism have not been tightly linked in the revised manuscript, although a speculating paragraph has been added to the discussion. Altogether, the study is rather descriptive, does not go beyond the current knowledge and provides limited novel insights into the underlying mechanisms. Overall, it does not qualify for publication to EMBO J.

Major concerns:

- In the first review, a serious concern was raised regarding the inconsistency of patient lines used. The authors addressed this by stating in the revised manuscript that they included "1 clone from each healthy subject (controls 1, 2, 3) and 3 from both patients p.F419fsX472a (#3, 5, 8) and b (# 3, 5, 11), and 1 from patient p.Y190X (#1). These different clones were subjected to the identical analysis, an example of which is reported in each figure." Moreover, they explain that they show the pooled results for all patients as "PKAN patients" compared to the pooled controls in several figures. Yet, this way of presenting the results does not display the variances between the lines generated from the different patients. Also unequal number of clones for the patients were used, thus the presentation of pooled data may lead to misrepresentation and consequent interpretation.

- There is significant inconsistency regarding the stage of differentiation during which electrophysiological recordings, mitochondrial mass, and cell survival have been assessed.

- The authors must address the stage of differentiation at which neuronal loss is observed. Are significant changes in neuron number at d 120 already present? This is important to evaluate the beneficial effect of CoA.

- In Figure 5B, information on the repetitively firing neurons with and without CoA should be given for both control and patient lines.

- "The integrity of mitochondrial membrane potential" cannot be assessed by TMRM staining without data on mitochondrial mass. In the response letter, the authors refer to the low yield of mitochondria in their cultures as a major drawback. Nevertheless, it is not needed to isolate mitochondria in order to assess the relative abundance of mitochondria. For instance, mitochondrial markers may also be detected in total lysates or MMP-insensitive dyes can be used. In any case, no inferences concerning mitochondria amount may be made by the observation of altered mitochondria morphology/size.

- The authors have already published (and they state it) that defects in mitochondrial activity, differences in oxidative status, and iron metabolism are observed in PKAN patient fibroblasts and neuronal cells generated by direct conversion. Although the authors include now electrophysiological studies and a possible therapeutic effect of CoA administration, their current model does not overcome the limitations of their previous models for clarifying the relationship between iron dysregulation, neuronal function, and cell death. Particularly, the mechanistic link between iron dysregulation, oxidative stress, mitochondrial dysfunction and neuronal function is still not clarified.

- The authors must clarify which patient lines and clones were used for electrophysiological recordings and mitochondria ultrastructural analysis.

Minor concerns:

- 1) Figure legends are extensive
- 2) Language still needs to be thoroughly revised in several parts of the manuscript.

Referee #2 (Comments on Novelty/Model System):

See below "Remarks to be sent to the author".

Referee #2 (Remarks):

The authors correctly answered to my questions and modified their manuscript in accordance except for one point:

Figure 4A. I understand that the measurement of aconitases in gel activity requires a large number of cells hampering to have a single gel with several controls and patients' cells. Nevertheless, the loading of PKAN(Y190X)#1 and the PKAN(F419fsX427a)#5 cells is not optimal and the background of the gel relatively high making the measurement of c.Aco activity not reliable on the

left gel. The right gel is much better. May be only this right gel should be used for aconitase quantification.

2nd Revision - authors' response

13 July 2016

Responses to Reviewer's comments

Below is a point-by-point reply to the referees' comments:

Referee #1

1- There is significant inconsistency regarding the stage of differentiation during which electrophysiological recordings, mitochondrial mass, and cell survival have been assessed.

We performed several electrophysiological recordings at different time points in the range between 4-20 weeks of differentiation. Cell survival has been evaluated on the same samples used for the recording and thus at same time points. However, to simplify the description of the data and for the sake of clarity, we previously showed only data at 1 and at 150 days (see Fig. 5A). In the revised version of the manuscript we also added survival data at 120 days, as required by referee (see point 2).

As concerning evaluation of mitochondrial mass, we followed the suggestion of the referee and tested the amount of the mitochondrial marker Tim44 (translocator of inner membrane) in total cellular lysates of neurons. We include the immunoblotting for referee's evaluation. As you can appreciate we did not detect any difference in Tim44 amount as compared with the loading control β -actin, thus indicating that mitochondrial content is comparable between control and PKAN neurons.

2- The authors must address the stage of differentiation at which neuronal loss is observed. Are significant changes in neuron number at d 120 already present? This is important to evaluate the beneficial effect of CoA.

As suggested by the referee, we added the neuron number at point 120d both in the text and in Figure 5A.

3- In Figure 5B, information on the repetitively firing neurons with and without CoA should be given for both control and patient lines.

In the previous revised version we added the number of repetitive firing control neurons with and without CoA treatment in the text. We have now added the same information also in Figure 5B.

4- The authors must clarify which patient lines and clones were used for electrophysiological recordings and mitochondria ultrastructural analysis.

We apologize for the lack of identification of the clones used for electrophysiological recordings. Now they have been added in Figures 1F and 5B. The numbers of clones used for mitochondria ultrastructural analysis were already reported in the corresponding picture.

Minor concerns:

5) *Figure legends are extensive*

All figure legends have been revised to eliminate duplication and redundant sentences.

6) *Language still needs to be thoroughly revised in several parts of the manuscript.*

An English mother tongue has revised the manuscript. His name has been added to the acknowledgements section.

Referee #2 (Remarks):

The authors correctly answered to my questions and modified their manuscript in accordance except for one point: Figure 4A. I understand that the measurement of aconitases in gel activity requires a large number of cells hampering to have a single gel with several controls and patients' cells. Nevertheless, the loading of PKAN(Y190X)#1 and the PKAN(F419fsX427a)#5 cells is not optimal and the background of the gel relatively high making the measurement of c.Aco activity not reliable on the left gel. The right gel is much better. May be only this right gel should be used for aconitase quantification.

We changed the left gel with another experiment presenting two controls and two patients in place of the previous one with all the three controls together. In the figure's legend we added a sentence reporting that the displayed gels are representative images of several independent experiments (at least three). Quantifications in the plot are the global results derived from all the performed experiments.

Corresponding Author Name: Sonia Levi
Journal Submitted to: EMBO Molecular Medicine
Manuscript Number: EMM-2016-06391